# Comparative Transcriptome Analysis Reveals the Molecular Basis of *Brassica napus* in Response to Aphid Stress

**DOI:** 10.3390/plants12152855

**Published:** 2023-08-03

**Authors:** Yuanhong Li, Lei Cai, Ting Ding, Entang Tian, Xiaohong Yan, Xiaodong Wang, Jiefu Zhang, Kunjiang Yu, Zhuo Chen

**Affiliations:** 1College of Agriculture, Guizhou University, Guiyang 550025, China; liyuanh31@163.com (Y.L.); caileixy@163.com (L.C.); 15008578109@163.com (T.D.); erictian121@163.com (E.T.); 2Center for Research and Development of Fine Chemical, Guizhou University, Guiyang 550025, China; 3Key Laboratory of Biology and Genetic Improvement of Oil Crops, Ministry of Agriculture and Rural Affairs, Oil Crops Research Institute, Chinese Academy of Agricultural Sciences, Wuhan 430062, China; yanxiaohong@caas.cn; 4Institute of Industrial Crops, Jiangsu Academy of Agricultural Sciences, Key Laboratory of Cotton and Rapeseed, Ministry of Agriculture and Rural Affairs, Nanjing 210014, China; xdwang120@163.com (X.W.); jiefu_z@163.com (J.Z.); 5Guangxi Tianyuan Biochemical Co., Ltd., Nanning 530009, China

**Keywords:** *Brassica napus*, resistance to aphid stress, transcriptome characteristics, physiological characteristics, candidate gene

## Abstract

Rapeseed is a globally important economic crop that can be severely impacted by aphids. However, our understanding of rapeseed resistance to aphid stress is very limited. In this study, we analyzed the resistance characteristics of the low aphid-susceptible variety APL01 and the highly aphid-susceptible variety Holly in response to aphid stress. APL01 had a more significant inhibitory effect on aphid proliferation compared with Holly during the early stage of inoculation, whereas Holly showed stronger tolerance to aphid stress compared with APL01 during the later stage of inoculation. Through transcriptome, physiological, and gene expression analyses, it was revealed that chitinase activity, catalase activity, calcium signal transduction, and activation of systemic acquired resistance might be involved in aphid resistance in *B. napus*. The degree of inhibition of photosynthesis in plants under aphid stress directly determines the tolerance of *B. napus* to aphid stress. Furthermore, four promising candidate genes were screened from eight genes related to rapeseed response to biotic stress through RT-qPCR analysis of gene expression levels. These research findings represent an important step forward in understanding the resistance of rapeseed to aphid stress and provide a solid foundation for the cloning of genes responsible for this resistance.

## 1. Introduction

Rapeseed *Brassica napus* (genome AACC, 2*n* = 38) is a highly profitable oil crop worldwide [1,2], the yield of which is significantly impacted by aphids [*Myzus persicae* (*Sulzer*)] and cabbage aphids [*Brevicoryne brassicae* (L.)] [3]. The use of chemical pesticides has proven highly successful in controlling aphid infestations in crops over the past few decades [4]. However, due to the increasing costs of these pesticides and the economic and environmental benefits associated with organic farming, this approach is now becoming less favored [4]. Therefore, the development of resistant crop varieties seems to be the most practical solution to address aphid problems [4]. Unfortunately, our limited knowledge about aphid resistance, defense, and tolerance in *B. napus* hinders the cultivation of genetically improved rapeseed varieties that are aphid-resistant, defensive, and tolerant.

Aphids are herbivores that mainly feed on the tender tissues of plants and harm organs, such as roots, stems, leaves, flowers, and fruits throughout the plant [5], mainly by sucking on tissues and by spreading viruses [5]. Aphids can produce and inject specific compounds (“effectors”) intended to modulate and suppress the phytohormonal and defensive response of susceptible plants [6]. However, plants have evolved mechanisms to respond to aphid stress. In resistant plants, aphid salivary compounds may be recognized by plants and activate targeted defenses, including the induction of plant secondary metabolites (PSMs) and other mechanisms of resistance [6]. Several types of PSM in phloem sap have been found to have defensive effects on aphid feeding, including terpenoids, alkaloids, anthocyanins, phenols, quinones, etc. [7,8,9,10]. Salicylic acid (SA), abscisic acid (ABA), and jasmonic acid (JA) play a key role in improving plant defense or tolerance to aphids [11,12,13]. Aphid infestation could induce the accumulation of SA in wheat, thereby stimulating defense responses [12]. In soybean, higher constitutive levels of ABA and JA and basal expression of ABA- and JA-related transcripts were found in the tolerant genotype [13]. Aphid effectors produced by the interaction between plants and aphids can trigger the production of reactive oxygen species (ROS), the deposition of callose in plant cells, and the induction of the defense response of plants [14]. ROS might be the first key signaling substance to integrate environmental information and regulate resistance to aphids [15]. In addition, sugars, such as sucrose, glucose, fructose, and trehalose, also play important roles in the signal transduction of plant aphid resistance regulation [16]. These substances can activate the release of intercellular chitinase and *β*-1,3-glucanase from oligosaccharides in the cell wall [16]. In addition, aphids can trigger plants to produce certain volatiles, which can not only affect the attractiveness of surrounding plants to aphids but also induce their resistance to these insects [17,18]. In addition to defensive PSMs, there are other changes in the phloem of plants that play a role in defending against aphid stress. Research has shown that an increase in the nutritional quality of phloem sap can reduce the rate of evolutionary change in aphids by increasing their population size and decreasing random drif [19]. To protect the valuable resources in their phloem sap from aphid feeding, plants have evolved various mechanisms [20]. One highly efficient mechanism is phloem sealing, which prevents aphid establishment but does not stop probing or the spread of plant viruses [21]. Phloem sealing is typically associated with other forms of defense, such as species-specific receptors for aphids [22]. Activation of pattern recognition receptors has been found to lead to higher rates of phloem sealing [22]. In the case of peach, aphids feeding on resistant cultivars showed increased probing and shorter feeding periods, indicating an increase in phloem sealing rates [23].

Regarding the genetic basis of plant response to aphid stress, genetic studies have shown that plant resistance, defense, and tolerance to aphids are jointly regulated by multiple loci and exhibit significant major locus/gene effects [24,25,26,27], with dominant [28], additive [29,30], and epistatic effects [30] among loci/genes. Researchers have identified at least ten genetic loci that are linked to aphid resistance in soybeans [24,30,31,32,33,34,35,36,37]; however, the specific molecular mechanisms through which these loci regulate soybean resistance to aphids remain unknown. In maize, two major loci associated with resistance to corn leaf aphids (*Rhopalosiphum maidis*) have been identified [29]. One of these loci potentially plays a role in DIMBOA (2,4-dihydroxy-7-methoxy-1,4-benzoxazin-3-one) biosynthesis and callose accumulation. [29]. One major and one minor genetic locus for aphid resistance have been identified in an African cowpea variety that was bred to be resistant to aphids [25]. In safflower, a locus responsible for aphid tolerance has been identified, explaining 31.5% of the variation in phenotype [26]. In rapeseed, there is a lack of genetic knowledge regarding plant response to aphid stress [2]. This lack of knowledge is one of the main reasons why there has been limited progress in breeding for resistance [2]. Although loci associated with plant resistance, defense, and tolerance to aphid stress have been identified in some plants, the genes related to these loci have rarely been cloned thus far. Three main types of aphid-resistance genes have been cloned thus far: aphid-resistant genes isolated from bacteria, such as the isopentenyl transferase (*IPT*) from *Agrobacterium tumefaciens* [38]; aphid-resistant genes isolated from plants, such as protease inhibitor genes [39]; and toxin genes isolated from animals, such as spider toxin genes [40]. Gene function research has found that aphid-resistant genes interact with nontoxic genes of aphids, leading to host plant resistance. However, this effect only occurs between specific plants and aphids [5], and these genes are specific aphid-resistant genes. In addition, some genes trigger effective resistance to different types or biotypes of aphids, being broad-spectrum aphid resistance genes, including mainly protease inhibitor genes [39,41], cytokinin synthesis-related genes [38], and plant lectin genes [42]. *CORONATINE INSENSITIVE 1* (*COI1*) is currently the only gene cloned from rapeseed that is associated with defense to aphids [43]. In *Arabidopsis*, *COI1*, as a core member of the jasmonate receptor complex, has a role in the jasmonate signaling pathway [44]. Silencing *COI1* expression in rapeseed weakens plant resistance to aphids [43]. However, the molecular mechanism by which *COI1* regulates aphid resistance in rapeseed is not clear; in addition, changes in its expression can affect male fertility in rapeseed [43], making it difficult to use for breeding aphid-resistant rapeseed.

In this study, two rapeseed varieties with different levels of susceptibility (APL01 with an average of 25% susceptibility and Holly with an average of 93% susceptibility from flowering to podding) to peach aphids (*Myzus persicae* (*Sulzer*)) were used for aphid defense, transcriptome, and physiological analyses. The objectives were to (1) evaluate whether there is a difference in defense against aphids between two *B. napus* varieties with different aphid susceptibility in the field and also determine the defense mechanisms of rapeseed against aphids (such as antibiosis or/and antixenosis) and (2) reveal the transcriptomic and physiological basis of rapeseed response to aphid stress. The molecular basis of differences in aphid stress response between APL01 and Holly will be revealed through this study. These findings help shed light on the molecular mechanism of rapeseed response to aphids and could be used to cultivate aphid-resistant varieties through molecular-assisted selection.

## 2. Results

### 2.1. Differences in Response to Aphid Stress between APL01 and Holly Varieties

To determine whether there are differences in the defense ability against aphids between APL01 and Holly, field investigations on aphid susceptibility and indoor aphid inoculation experiments were conducted. Field observations showed that the aphid density on APL01 was significantly less than that on Holly during the flowering and podding stages (Figure 1). From flowering to podding, the susceptibility to aphids in APL01 is significantly lower at 25% compared to Holly’s high susceptibility of 93% (Appendix A). The indoor aphid inoculation experiment showed that the number of aphids on APL01 and Holly increased from the 7th day after inoculation (Table 1, Appendix A). Except for the 7th and 13th days, the number of aphids on Holly was significantly higher than that on APL01. From the 13th to the 16th day, the number of aphids on Holly increased more quickly compared with the number on APL01 (Table 1, Appendix A). These results suggest that APL01 has a stronger inhibitory effect on aphid proliferation (i.e., defense) compared with Holly.

Unexpectedly, 16 days after inoculation with aphids, the number of aphids on APL01 and Holly increased significantly, which made it difficult to count them. Nevertheless, observations of the phenotypes of APL01 and Holly continued until the 25th day after aphid inoculation. The results showed that most APL01 plants withered, with a withering rate of 66.7% compared with 28.6% in Holly (Table 1, Appendix A), indicating that Holly has stronger tolerance to aphid stress than APL01.

To investigate potential differences in the toxic effects on aphids (i.e., antibiosis) between APL01 and Holly varieties, toxicity tests were performed. Leaf extracts of APL01 and Holly and blank control (the extract solution without any powdered leaf added) were used to soak young leaves of Chinese cabbage for 1 h and then inoculated with aphids. After 24 h of feeding, the mortality of aphids on the leaves soaked with APL01 extract, Holly extract, and blank control was 68.3%, 63.3%, and 6.7%, respectively (Table 2), indicating that APL01 and Holly have a certain degree of toxicity (i.e., antibiosis) to aphids.

### 2.2. Transcriptomic Analysis of APL01 and Holly in Response to Aphid Stress

To elucidate the transcriptomic basis of the differential response to aphid infestation between APL01 and Holly, the young leaves of APL01 inoculated with aphids for 13 days (referred to as IA) and non-inoculated (referred to as NIA), as well as the young leaves of Holly inoculated with aphids for 13 days (referred to as IH) and non-inoculated (referred to as NIH), were collected for RNA sequencing. Comparative transcriptomic analysis revealed 1343 genes that were significantly upregulated in IA compared with NIA, whereas 548 genes were downregulated (Table 3); 131 genes were significantly upregulated in IH compared with NIH, whereas 71 genes were downregulated (Table 3); 3187 genes were significantly upregulated in IA compared with IH, whereas 2540 genes were downregulated (Table 3); 2255 genes were significantly upregulated in NIA compared with NIH, whereas 2217 genes were downregulated (Table 3). Subsequently, Venn analysis, Gene Ontology (GO), Kyoto Encyclopedia of Genes and Genomes (KEGG), and enrichment analyses of these differentially expressed genes (DEGs) were carried out to reveal the transcriptomic characteristics of APL01 and Holly in response to aphid stress.

### 2.3. Common Transcriptomic Characteristics of APL01 and Holly Involved in Aphid Stress Defense

To reveal the common transcriptomic characteristics of APL01 and Holly against aphids, a Venn analysis between IA vs. NIA and IH vs. NIH groups was conducted. In total, 74 DEGs were identified to be upregulated in common in the IA vs. NIA and IH vs. NIH groups (Figure 2A).

GO enrichment analysis showed that these DEGs were significantly enriched in 36 terms related to biological processes, seven terms related to cellular components, and 24 terms related to molecular functions (Appendix A). The hierarchical relationship between these significantly enriched GO terms was shown in a topGO-directed acyclic graph (DAG) (Appendix A). The results showed that 36 terms related to biological processes were ultimately subdivided into a specific term for the functional description of the “hydrogen peroxide catalytic process” (GO:0042744) (Figure 3, Appendix A). Seven terms related to cellular components were subdivided into three terms with specific functional descriptions of “CCR4-NOT complex” (GO:0030014), “actomyosin contractual ring” (GO:0005826), and “core action cyclone” (GO:0030864) (Figure 3, Appendix A). The 24 terms related to molecular functions were subdivided into four terms with specific functional descriptions of “poly (A)-specific ribonuclease activity” (GO:0004535), “calcium ion binding” (GO:0005509), “ubiquitin–protein ligase activity” (GO:0061630), and “peroxidase activity” (GO:0004601) (Figure 3, Appendix A). Two of the eight GO terms with the most specific functional descriptions involved hydrogen peroxide catabolism; two involved RNA degradation; two involved leaf stomatal responses to pathogen invasion; one involved ubiquitin protein degradation pathway, and one involved cell calcium signal transmission. All these processes have a role in plant defense against alien biological invasion.

In addition, KEGG enrichment analysis showed that the 74 DEGs above were significantly enriched only in the pathway of “RNA degradation” (ko3018) (Appendix A). Several previous studies have shown that the regulation of mRNA homeostasis has an important role in plant defense against bacterial [45], fungal [46], and viral [47] attacks. Therefore, it cannot be ruled out that mRNA degradation was involved in the of *B. napus* to aphid stress in this study.

### 2.4. Transcriptomic Characteristics of APL01 with Stronger Inhibitory Effects on Aphid Proliferation Compared with Holly

To reveal the transcriptomic basis of the stronger inhibition of APL01 on aphid proliferation compared with Holly, DEGs that were uniquely upregulated in the IA vs. NIA group and also upregulated in the IA vs. NIA, IH vs. NIH, and IA vs. IH groups were identified (Figure 2A).

GO enrichment analysis of 1269 DEGs uniquely upregulated in the IA vs. NIA group found that these DEGs were significantly enriched in 232 terms related to biological processes, 22 terms related to cellular components, and 108 terms related to molecular functions (Appendix A). Furthermore, DAG showed that these 362 terms subdivided into 14 terms with specific functional descriptions (Figure 4, Appendix A), three of which involved chitin catabolism [“chitin catabolic process” (GO:0006032), “chitin binding” (GO:0008061), and “chitinase activity” (GO:0004568)]. Coincidentally, the GO enrichment analysis of 1583 DEGs commonly upregulated in the IA vs. IH and NIA vs. NIH groups showed that these DEGs were enriched in 13 terms with specific functional descriptions (Appendix A, Appendix A, Appendix A), one of which was related to chitin binding (Appendix A). All of these results suggest that chitin decomposition ability has an important role in the resistance of APL01 to aphid stress. The other three terms, “systematic acquired resistance” (GO:0009627), “cell wall macromolecule catabolic process” (GO:0016998), and “calcium ion binding” (GO:0005509), also relate to the plant immune response to biotic stress (Figure 4). It cannot be ruled out that these terms also contribute to the resistance of APL01 to aphid stress in this study. In addition, KEGG enrichment analysis showed that these 1269 DEGs were significantly enriched in 17 pathways, 12 of which are related to plant defense against biotic or abiotic stresses, including “Plant–pathogen interaction”, “Phenylpropanoid biosynthesis”, “Glucosinolate biosynthesis”, “Glutathione metabolism”, “Phosphatidylinositol signaling system”, and “Plant hormone signal transduction” (Appendix A).

Venn analysis identified 13 genes that were commonly upregulated in the IA vs. NIA, IH vs. NIH, and IA vs. IH groups (Figure 2A). GO enrichment analysis showed that these 13 DEGs were significantly enriched in 18 terms related to biological processes and 13 terms related to molecular functions (Appendix A). Based on DAG, these 31 terms were subdivided into seven terms with specific functional descriptions (Figure 5, Appendix A), five of which were consistent with those terms that were significantly enriched by the unique upregulated DEGs in the IA vs. NIA group: “cell wall macromolecule catabolic process” (GO:0016998), “chitin catabolic process” (GO:0006032), “calcium ion binding” (GO:0005509), “chitin binding” (GO:0008061) and “chitinase activity” (GO:0004568). This shows that APL01 has a stronger chitin decomposition ability, cell wall defense ability, and cell defense signal transduction efficiency compared with Holly in resistance to aphid stress. KEGG enrichment analysis showed that these 13 DEGs were only enriched in one pathway, “Ubiquinone and another terpenoid–quinone biosynthesis” (ko130) (Appendix A). Previous studies reported that naphthoquinones and anthraquinones in plants have a role in preventing pests [48,49]; thus, the quinone synthesis in this study might also be related to the stronger defense of APL01 to aphid stress compared with Holly.

Further analysis of the expression of genes related to plant resistance or defense to biotic stress showed that except for the genes related to JA and SA signal transduction pathways that were uniquely upregulated only in the IH vs. NIH group, other resistance or defense genes were mainly upregulated in the IA vs. NIA group, with the number and proportion of upregulated genes being higher than in the IH vs. NIH group (Table 4). This not only explains why APL01 and Holly both have inhibitory effects on aphid proliferation but also clarifies the transcriptomic basis that APL01 has stronger inhibitory effects than Holly.

### 2.5. Common Transcriptomic Characteristics of Growth Inhibition in APL01 and Holly under Aphid Stress

To reveal the adverse effects of aphid stress on the normal growth of *B. napus*, 26 DEGs that were commonly downregulated in the IA vs. NIA and IH vs. NIH groups were identified (Figure 2B). GO enrichment analysis showed that these 26 DEGs were significantly enriched in 29 terms related to biological processes, 32 terms related to cellular components, and 18 terms related to molecular functions (Appendix A). The DAG shows that these 79 terms were subdivided into 12 terms with specific functional descriptions (Appendix A), of which seven are related to plant photosynthesis (Figure 6): “photosynthesis, light harvesting” (GO:0009765), “protein–chromophore linkage” (GO:0018298), “photosystem II” (GO:0009523), “photosystem I” (GO:0009522), “chloroplast thylakoid membrane” (GO:0009535), “chlorophyll binding” (GO:0016168), and “protein serine/threonine phosphatase activity” (GO:0004722). Furthermore, KEGG enrichment analysis of these 26 DEGs also showed that they were only enriched in two pathways related to photosynthesis: “Photosynthesis—antenna proteins” (ko196) and “Carotenoid biosynthesis” (ko906) (Appendix A). These results indicated that the photosynthesis of APL01 and Holly was inhibited after aphid stress, which could also explain why APL01 and Holly plants withered after 25 days of aphid stress.

### 2.6. Transcriptomic Characteristics of the Stronger Tolerance of Holly to Aphid Stress Compared with APL01

To reveal the transcriptomic basis of the stronger tolerance of Holly than of APL01 after 25 days of aphid stress, DEGs that were uniquely downregulated in the IA vs. NIA or the IH vs. NIH groups and that were commonly downregulated in the IA vs. NIA, IH vs. NIH, and IA vs. IH groups were identified (Figure 2B).

GO enrichment analysis showed that 522 DEGs uniquely downregulated in the IA vs. NIA group were mainly enriched in 203 terms related to biological processes, 62 terms related to cellular components, and 90 terms related to molecular functions (Appendix A). The DAG showed that these 355 terms are subdivided into 14 terms with specific functional descriptions, all of which involve plant photosynthesis (Figure 7, Appendix A). KEGG enrichment analysis of these 522 DEGs showed that they were significantly enriched in 16 pathways mainly involved in biological processes, such as photosynthesis, gluconeogenesis, and amino acid metabolism (Appendix A). These results indicate that more genes involved in photosynthesis in APL01 are downregulated following aphid stress, potentially leading to severe obstruction of photosynthesis in APL01.

Venn analysis of the IA vs. NIA, IH vs. NIH, and IA vs. IH groups identified four DEGs that were downregulated in all three groups (Figure 2B). GO enrichment analysis found that these four DEGs were significantly enriched in seven terms related to biological processes, 20 terms related to cell components, and ten terms related to molecular functions (Appendix A). DAG showed that these 37 terms were subdivided into ten terms with specific functional descriptions (Appendix A), nine of which were involved in the regulation of plant photosynthesis (Figure 8). KEGG enrichment analysis showed that these four DEGs were only enriched in “Photosynthesis—antenna proteins” (ko196), a pathway related to photosynthesis (Appendix A). This indicates that photosynthesis in APL01 is more seriously blocked than in Holly under aphid stress.

GO enrichment analysis of 45 DEGs uniquely downregulated in the IH vs. NIH group showed that these DEGs were mainly enriched in 102 GO terms (Appendix A). Further combining these results with those of DAG, these terms are subdivided into three terms related to biological processes, three terms related to cellular components, and six terms related to molecular functions (Appendix A). Of the 12 terms with specific functional descriptions, six are related to photosynthesis (Figure 9). KEGG enrichment analysis also showed that these 45 DEGs were significantly enriched in pathways related to photosynthesis (Appendix A). In addition to the terms related to photosynthesis, DEGs were also found to be significantly enriched in the “unsaturated fatty acid biosynthetic process” (GO:0006636) (Figure 9). Given that unsaturated fatty acids might be the food of aphids, as it is for *Caenorhabditis elegans* and *Musca domestica* larvae [50,51], the unique downregulated expression of genes related to unsaturated fatty acid biosynthesis in the IH vs. NIH group may be related to the tolerance of Holly to aphid stress.

In addition, GO enrichment analysis of 1566 DEGs commonly downregulated in the NIA vs. NIH and IA vs. IH groups showed that 15 terms with specific functional descriptions included “cell redox homeostasis” (GO:0045454) (Appendix A). A previous review showed that cell redox homeostasis has an important role in the regulation of electron transport at the plasma membrane and is necessary for cell adaptation and normal respiration and photosynthesis [52]. Therefore, it is likely that Holly has better cellular redox homeostasis, which contributes to the stronger tolerance of these plants to aphid stress compared with APL01.

Further analysis of the expression levels of genes related to plant photosynthesis showed that numerous genes were significantly downregulated in the IA vs. NIA group, with fewer and a smaller proportion of downregulated genes in the IH vs. NIH group (Table 5). Thus, the main reason why APL01 is more prone to wilting compared with Holly under aphid stress is that photosynthesis in the former is more severely damaged by aphids than in the latter. However, it is unclear why photosynthesis in Holly is less disrupted than in APL01 after aphid stress.

### 2.7. Physiological Characteristics of APL01 and Holly in Response to Aphid Stress

The above transcriptome analysis results indicated that ROS decomposition is potentially related to the response of *B. napus* to aphid stress (Table 4, Figure 3). To verify this further, we measured the activities of catalase (CAT) and peroxidase (POD) in APL01 and Holly leaves at different times after inoculation with aphids. The comparative analysis showed that the POD activity in APL01 leaves continued to increase from before inoculating aphids to the 13th day after inoculation, whereas it began to decrease after 7 days of inoculation in Holly (Table 6). The activity of CAT first increased and then decreased in both APL01 and Holly after inoculation with aphids, although the increase in APL01 was greater than that in Holly and the decrease smaller than that in Holly (Table 6). These results potentially reflect a greater increase in ROS content in APL01 leaves compared with Holly after aphid stress, thereby inducing a faster and stronger defense response in the former.

Further analysis revealed that, before inoculation with aphids, chitinase activity in APL01 leaves was significantly higher than that of Holly, whereas, on days 7 and 13 of aphid inoculation, there was no significant difference in chitinase activity between the two varieties (Table 6). Given that chitinase might be involved in chitin decomposition in insect epidermis, this might be one reason why both APL01 and Holly leaf extracts have toxic effects on aphids, although the difference in toxic effects between the two varieties is not significant. The difference in aphid proliferation rates between APL01 and Holly might be due to other factors, such as calcium ion signal transduction and systemic acquired resistance activation. These factors could promote the synthesis of certain PSMs in rapeseed, resulting in plant antixenosis to aphids.

In addition, the results of the analysis of plant tolerance to aphid stress and transcriptome analysis potentially showed that the photosynthesis of APL01 is blocked more significantly than in Holly after aphid stress (Table 1 and Table 5, Appendix A). We also analyzed the net photosynthetic rate of APL01 and Holly before and after inoculation with aphids, showing that the net photosynthetic rate of APL01 increased first and then decreased with inoculation time, whereas that in Holly continued to increase with inoculation time, with its net photosynthetic rate being significantly higher than APL01 at different times after inoculation (Table 7). In terms of the chlorophyll content, on the 13th day after aphid inoculation, the relative chlorophyll content in APL01 leaves significantly decreased compared with before inoculation, whereas the decrease in Holly was smaller than that in APL01 (Table 7). Furthermore, we analyzed the activities of two key enzymes involved in photosynthesis, showing that the activities of ribulose-bisphosphate carboxylase (Rubisco) and fructose-bisphosphate aldolase (FBA) continued to decrease in APL01 after inoculation with aphids, but continued to increase in Holly (Table 7). All these results indicate that during the early stages of aphid stress (0–13 days after inoculation), APL01 photosynthesis is severely hindered, whereas that of Holly is less affected.

### 2.8. Screening of Candidate Genes Related to the Rapeseed Defense to Aphid Stress

The above transcriptome and physiological analysis results show that chitinase activity, CAT activity, calcium signal transduction, and activation of systemic acquired resistance are potentially involved in aphid defense in *B. napus*. We used quantitative real-time reverse transcription PCR (RT-qPCR) to analyze the expression of eight DEGs closely related to these GO terms to screen candidate genes for aphid defense in rapeseed.

Analysis of the expression levels of two genes related to chitin binding, BnaC03g37570D and BnaC03g37600D, showed that their expression levels in APL01 were significantly higher than those in Holly before and on the 13th day after aphid inoculation (Figure 10). However, the expression level of BnaC03g 37570D in APL01 and Holly did not significantly change from before aphid inoculation to the 13th day after inoculation (Appendix A), which is consistent with the results of chitinase activity analysis (Table 6). Therefore, BnaC03g37570D is considered a potential candidate gene that might be involved in the antibiosis of rapeseed to aphids by promoting chitin decomposition in the aphid exoskeleton.

The two genes related to hydrogen peroxide decomposition metabolism, BnaAnng32690D and BnaC09g25420D, showed significantly higher expression levels in APL01 than in Holly before aphid inoculation (Figure 10). However, compared with the expression level before aphid inoculation, both genes were significantly downregulated in APL01 on the 13th day after aphid inoculation, whereas no significant changes were observed in Holly on the 13th day after aphid inoculation (Appendix A). The expression patterns of these two genes in APL01 and Holly from before aphid inoculation to the 13th day after inoculation were not consistent with the changes in POD and CAT activity in these two varieties. Therefore, they are not considered candidate genes.

Analysis of the expression levels of genes related to calcium ion binding showed no significant difference in the expression levels of BnaA01g16310D and BnaC04g47650D between APL01 and Holly before aphid inoculation (Figure 10). However, after 13 days of inoculation, the expression levels of both genes were significantly higher in APL01 than in Holly (Figure 10). In terms of gene expression patterns, both BnaA01g16310D and BnaC04g47650D were significantly downregulated in Holly on the 13th day after aphid inoculation compared with before inoculation, whereas only the expression level of BnaA01g16310D was downregulated in APL01 on the 13th day after inoculation, with BnaC04g47650D being slightly upregulated (Appendix A). Therefore, it is likely that BnaC04g47650D has an important role in calcium signaling transduction in APL01 after 13 days of aphid inoculation, contributing to its stronger defense against aphid stress compared with Holly.

Of the two genes associated with the regulation of systemic acquired resistance, BnaA03g11410D was significantly more highly expressed in APL01 than in Holly before aphid inoculation, but its expression level in Holly was significantly higher than in APL01 on the 13th day post-aphid inoculation (Figure 10). Based on the gene expression pattern over time, the expression level of BnaA03g11410D in Holly was significantly upregulated with increasing aphid inoculation time, whereas there was no significant change in expression level in APL01 (Appendix A). Additionally, the expression level of BnaC02g13020D in APL01 was significantly lower than in Holly before inoculation and on the 13th day post-aphid inoculation (Figure 10 and Appendix A). Based on these results, it is likely that BnaA03g11410D and BnaC02g13020D are the candidate genes for regulating the defense (i.e., antibiosis) of Holly to aphid stress.

In addition, the expression levels of two genes involved in carbon fixation, two genes involved in the regulation of key enzyme activity in photosynthesis, and two genes involved in light harvesting were also analyzed (Appendix A). Their expression levels in APL01 decreased sharply with increasing aphid inoculation time, whereas their expression levels in Holly did not (except for the two genes involved in light harvesting) (Appendix A). This further reflects that photosynthesis in APL01 was more severely inhibited at the gene expression level than in Holly after aphid stress.

## 3. Discussion

### 3.1. Antibiosis, Antixenosis, and Tolerance all Have a Role in the Response of Brassica napus to Aphid Stress

There are three functions involved in plant–pest interactions [53]: antibiosis; antixenosis; and tolerance. In different plants, antibiosis and/or antixenosis and/or tolerance may individually or collectively counteract insect stress [21]. The results of this study showed that both APL01 and Holly had a significant and similar toxic effect (i.e., antibiosis) on aphids. This suggests that the difference in defense to aphids between APL01 and Holly can also be attributed to the antixenosis of certain PSMs to aphids, although it is unknown which PSMs. Further experiments indicate that Holly has stronger tolerance than APL01 under aphid stress. In conclusion, different rapeseed varieties have different response mechanisms to aphid stress. APL01 mainly relies on antibiosis and antixenosis to cope with aphid stress, while Holly primarily uses antibiosis and tolerance to respond to aphid stress.

### 3.2. Potential Correlation between the Enzyme Activity of Scavenging ROS and the Defense of Brassica napus to Aphid Stress

Oxygen is usually used as the electron acceptor of plants in vivo [54]. After capturing electrons, oxygen generates superoxide anion O_2_^−^ and its derivatives H_2_O_2_·OH^−^, molecular oxygen, and other free radicals [54]. In general, ROS generated by cell metabolism does not harm plants; however, when plants are stressed by pests, numerous ROS will be produced, causing an imbalance of the ROS system while inducing the defense response of plants [14]. Plants can eliminate redundant active oxygen substances through coordination of CAT and POD activities to prevent the damage of active oxygen free radicals to the body and maintain the normal level of free radicals in cells to balance the active oxygen system [55,56]. This study demonstrates that the increase in POD and CAT activities in APL01 after inoculation with aphids is significantly higher than in Holly. These results might indirectly reflect a greater increase in ROS content in APL01 compared with Holly after aphid stress. Compared with Holly, the more significant increase in ROS content in APL01 leaves under aphid stress might lead to an earlier and faster defense response of APL01 to aphids during the early stages (within 16 days after inoculation) of aphid stress.

### 3.3. Role of Chitinase Activity in the Antibiosis of Brassica napus to Aphid

Plant chitinase is a disease-related protein that can destroy the cell wall of pathogenic bacteria and has an important role in plant disease resistance [57,58,59]. The peritrophic membrane, exoskeleton, and skin of nematodes can also be degraded by plant chitinase; thus, chitinase participates in the defense response of plants to pests [57]. Plant chitinase activity is directly regulated by biological stresses such as fungi and insects [59], as well as by disease resistance-related molecules, such as salicylic acid and ethylene [58]. Plant chitinase with an insect-resistance function has only been reported in carnivorous plants, such as *Drosera rotundifolia* [60]. In the current study, a large number of genes related to chitinase activity, chitin-binding, and chitin catabolism were found to be significantly upregulated in APL01 and Holly inoculated with aphids. However, the chitinase activity in APL01 leaves was comparable to that in Holly at different times after inoculation with aphids, which may explain why APL01 and Holly show similar antibiosis to aphids. Considering the potential role of chitinase activity in APL01 and Holly’s antibiosis to aphids, it can not be ruled out that APL01 and Holly may have a certain toxic effect on other piercing–sucking insects. However, this effect is likely limited to these types of insects and may not affect chewing insects due to the relatively low chitinase activity exhibited by these two varieties. Additionally, APL01’s antixenosis against aphids may also confer protection against other piercing–sucking insects.

### 3.4. The Degree of Inhibition of Photosynthesis under Aphid Stress Directly Determines the Tolerance of Brassica napus to Aphid Stress

Aphids harm plants mainly in two ways [3] by sucking plant sap or releasing toxic substances, affecting the normal physiological and metabolic activities and growth and development of plants; by contrast, as vectors of plant pathogens, aphids harm plants by spreading viruses. Other studies have shown that honeydew on the surface of plants by aphids could significantly affect the plant respiration and photosynthesis [61,62]. In this study, transcriptomic analysis showed that the number of downregulated genes related to photosynthesis in the IA vs. NIA group was higher than in the IH vs. NIH group. Physiological analysis showed that all measured photosynthetic indicators sharply decreased in APL01 after aphid inoculation in comparison with a small degree of increase or decrease in Holly. These results indicate that the degree of inhibition of photosynthesis in Holly after aphid stress was lower than in APL01, which might be why Holly showed a stronger tolerance to aphid stress compared with APL01. Previous studies have shown that plants could improve their stress tolerance by enhancing photosynthesis, such as increasing the content of total chlorophyll and carotenoids after being stressed by aphids [63,64]. Therefore, the degree of inhibition of photosynthesis under aphid stress directly determines the tolerance of *B. napus* to this stress.

In addition, GO enrichment analysis of 57 DEGs uniquely upregulated in the IH vs. NIH group suggested that JA and SA signal transduction pathways have a key role in the response of Holly to aphids (Appendix A). In fact, generalized plant responses to aphid feeding are mediated by phytohormonal signaling [21]. The signal pathways of JA and SA have been shown to have important roles in the tolerance of other plants to insects [65,66,67,68,69]. From this, it can be inferred that compared to APL01, Holly exhibits stronger tolerance to aphid stress, which may be related to the JA and SA signaling pathways.

### 3.5. Other Factors Potentially Related to the Defense of Brassica napus to Aphid Stress

In addition to hydrogen peroxide and chitin decomposition-related genes, other genes related to calcium ion binding, ubiquitin–protein binding, systemic acquired resistance, and cell wall macromolecule decomposition were upregulated in APL01 and Holly inoculated with aphids compared with those not inoculated. This might indicate that these genes are also involved in the defense of APL01 and Holly to aphid stress. The number of these genes in the IA vs. NIA group was higher than in the IH vs. NIH group, which might also explain why APL01 has a stronger defense against aphid stress compared with Holly. Previous studies have shown that Ca^2+^ homeostasis has an important role in plant immunity and cell survival [70]. The ubiquitination degradation of proteins has a decisive role in jasmonate signal release and activation of insect resistance response in plants [71,72]. Plants can trigger systemic acquired resistance after receiving external stimuli to protect their distal tissues from subsequent attacks by broad-spectrum pathogens [73,74,75]. In addition, increasing evidence also shows that the plant cell wall is not only a passive physical defense barrier but also a dynamic active defense system [76]. During pathogen infection, the integrity of the cell wall is often directly destroyed [76]. After plants perceive cell wall damage, they usually trigger a series of defense responses, including changes in the chemical composition and structure of the plant cell wall [76].

### 3.6. Potential Correlation between Host Plant Defense against Chewing Insects and Piercing–Sucking Insects

Compared to chewing herbivores, aphid feeding typically causes relatively little harm to a plant [21]. However, herbivores that chew on leaves and aphids that cause damage to plant cells both trigger the synthesis of JA and SA in a wide range of plant species [77,78]. This activation plays a critical role in facilitating the broad plant responses to insect feeding [77,78]. Considering that Holly shows a more sensitive response to JA and SA under aphid infestation compared to APL01, it can be inferred that Holly also has a stronger tolerance to chewing insect stress.

PSMs play a key role in resistance against many chewing herbivores, but their role against aphids is less clear [21]. Nevertheless, it has been observed that aphids cause structural damage to plants, which, in turn, activates enzymes that release defensive compounds from inactive forms [21]. This study demonstrates that APL01 may have a stronger defense mechanism against aphid stress due to its antixenotic effects on aphids. It is possible that APL01 also possesses some degree of antixenosis against specific chewing insects.

## 4. Materials and Methods

### 4.1. Plant Material and Aphid Inoculation Experiment

Two *B. napus* varieties (APL01 and Holly) with significantly different aphid susceptibility and aphid density on plants in the field were used as materials for this study (Appendix A). These two varieties were planted in adjacent plots of the Guizhou University field (Huaxi District, Guiyang City, Guizhou Province). Each plot consisted of 10 rows, with 10 plants in each row. Each row measured 0.15 m × 0.4 m in size. All 100 plants of each variety were selected for aphid infestation during the flowering and podding stages in the field. The two varieties were cultured in a climate-controlled box, with a light temperature of 25 °C for 16 h and a dark temperature of 20 °C for 8 h. The relative humidity was maintained at 80%. Seedlings with 4 or 5 fully developed leaves were used to inoculate aphids.

Peach aphids [*Myzus persicae* (*Sulzer*)] were collected from the experimental rapeseed field of Guizhou University. The aphids were inoculated on the young leaves of Chinese cabbage (the material that the peach aphids prefer to feed on) and raised in an artificial climate box (temperature 24 °C, relative humidity 72%, 16 h light/8 h dark). Aphids were used for inoculation experiments after three generations of aphid reproduction and purification. After 4 h of starvation, second-instar peach aphid nymphs were inoculated on the top tender leaves (the first fully developed leaf at the top of each plant) of APL01 and Holly according to the method of Deng et al. [79]. The aphid inoculation experiment included two treatments, inoculation and non-inoculation, with the single factor of variety (APL01 and Holly). Considering the extremely fast reproduction speed of peach aphids, only five aphids were added for each of the three plants of each variety in case the counting of aphids at different times after inoculation was accurate. After inoculation, each plant was immediately covered with an isolation cover to prevent aphids from transferring to other plants. Plants that were not inoculated with aphids were also covered with the same isolation cover to control for differences in environmental factors affecting plant growth, such as light, temperature, and humidity, between the inoculation and non-inoculation treatments. All plants, regardless of treatment, were cultured in an artificial climate box with controlled conditions (light 25 °C, 16 h; dark 20 °C, 8 h; relative humidity 80%). The number of aphids on each plant was then counted on the 7th, 10th, 13th, and 16th days after inoculation. After 16 days of inoculation, the number of aphids increased dramatically, and it was difficult to count them, so only the phenotype of each plant was observed. The calculation of the mean and standard deviation, as well as the comparison between means, was performed using SAS v9.4 software. The unpaired *t*-test method was employed to analyze the significance of differences between pairwise data.

### 4.2. Toxicity Test

Young leaves (the first fully developed leaf at the top of each of three plants per variety) from APL01 and Holly at the same developmental stage were removed, washed with sterile water, and then dried on filter paper. Then, 10 g of leaves were placed in a mortar and ground to a powder. Next, 4 mL of 95% ethanol was added, and the mixture was shaken thoroughly for 48 h. A filter was then used to remove the residue from the leaves, and then the filtrate was diluted to 20 mL with double distilled water, resulting in the leaf extract. The extract solution without any powdered leaf added was used as a control (i.e., blank extract). The young leaves (the first fully developed leaf at the top of each of three plants per variety) of Chinese cabbage were soaked in fresh leaf extract and blank control solution for 1 h, and any excess extract was then removed using absorbent paper. The leaves were placed in a culture dish and inoculated with 20 aphids (starved for 4 h). The culture dishes were placed in an artificial climate chamber, and the number of dead aphids was counted 24 h after inoculation. The toxicity test was repeated three times for each rapeseed variety. The significance of the difference between pairwise data was analyzed using the unpaired *t*-test method in SAS v9.4 software.

### 4.3. RNA Sequencing and Bioinformatics Analysis

Young APL01 leaves (the first fully developed leaf at the top of each plant) at the same developmental stage that were not inoculated with aphids (NIA) and those that had been inoculated with aphids for 13 days (IA) were collected. Similarly, young Holly leaves (the first fully developed leaf at the top of each plant) at the same developmental stage that were not inoculated with aphids (NIH) and those that had been inoculated with aphids for 13 days (IH) were collected. The leaves from three plants in each treatment group (i.e., IA vs. NIA and IH vs. NIH) of each variety (APL01 or Holly) were collected and combined into a biological sample, with each treatment containing three biological samples, resulting in 12 biological samples of the two varieties for RNA sequencing.

Total RNA was isolated using a MagaZorb^®^ Total RNA Mini-Prep Kit (Promega, Madison, WI, USA), whereas RNA degradation and contamination were monitored on 1% agarose gels. The purity of the RNA was checked using a NanoPhotometer^®^ spectrophotometer (IMPLEN, Westlake Village, CA, USA), and RNA concentration was measured using a Qubit RNA Assay Kit with a Qubit 2.0 Fluorometer (Life Technologies, Carlsbad, CA, USA). mRNA library construction, sequencing, and raw data processing were performed by Baimaike Biotechnology Co. (Beijing, China). An Illumina Novaseq™ 6000 (Illumina, CA, USA) was used for RNA sequencing, with sequencing read lengths of 150 bp with paired ends. For further details on constructing mRNA sequencing libraries and data analysis methods, please refer to Feng et al. [80].

The “Darmor-*bzh*” genome (www.genoscope.cns.fr/brassicanapus/) (accessed on 1 August 2022) was used as the reference genome for *B. napus* [1]. A false discovery rate of ≤0.05 and an absolute value of log_2_(Fold Change) ≥ 2 were used as the criteria for screening DEGs. GO enrichment analysis of the DEGs was implemented using the GOseq R package [81]. KEGG enrichment analysis of the DEGs was implemented using a hypergeometric distribution test as used previously [82]. A Kolmogorov–Smirnov (KS) value of ≤0.05 and a *q* value of ≤0.05 were used as criteria for determining significant GO and KEGG enrichment, respectively. For the significantly enriched GO terms, a DAG was drawn using the topGO R package to illuminate the relationships among these terms [83]. The GO and KEGG enrichment analyses of DEGs were both performed using the Baimaike Biological Cloud Platform (https://international.biocloud.net/zh/dashboard) (accessed on 1 October 2022).

### 4.4. Determination of Photosynthetic Characteristics

The net photosynthetic rate and relative chlorophyll content of APL01 and Holly leaves were measured using a portable photosynthetic measurement system equipped with a red and blue light source leaf chamber (Li-6400XT, LI-COR Biosciences, Lincoln, NE, USA) and SPAD (SPAD-502Plus, Konica Minolta, Tokyo, Japan), respectively. The net photosynthetic rate of the first fully developed leaf at the top of each selected plant was measured before inoculation as well as 7 and 13 days after inoculation with aphids. Additionally, the relative chlorophyll content at three positions in the distal third of the first fully developed leaf at the top of each selected plant was measured between 9:00 and 11:00 am before inoculation and 7 and 13 days after inoculation with aphids.

### 4.5. Determination of Peroxidase, Catalase, and Chitinase Activities

The POD and CAT activities of APL01 and Holly were measured using the Micro Peroxidase Assay Kit (Solarbio, Beijing, China) and Catalase Assay Kit (Solarbio), respectively. Ultraviolet (UV) spectrophotometry was also used to measure the activity of these two enzymes. For more details on POD activity assay methods, please refer to Wang et al. [84] and, for CAT activity assay methods, please refer to Zhang et al. [85]. The chitinase activity of APL01 and Holly was measured by visible spectrophotometry using the Chitinase Assay Kit (Solarbio), following the manufacturer’s instructions. The activity of POD, CAT, and chitinase enzymes in each variety was measured before, as well as 7 and 13 days after inoculation with aphids. Three plants per variety were selected for each treatment (before inoculation, 7 days after inoculation, and 13 days after inoculation with aphids). The first fully developed leaf at the top of each plant was collected and mixed to create a biological sample. Each treatment group consisted of three biological samples.

### 4.6. Determination of Ribulose-Bisphosphate Carboxylase and Fructose-Bisphosphate Aldolase Activities

The Rubisco and FBA activities of APL01 and Holly were assessed by UV spectrophotometry using the Rubisco Assay Kit (Solarbio) and the FBA Assay Kit (Solarbio), respectively, following the manufacturer’s instructions. The Rubisco and FBA activities of each variety were measured before inoculation and 7 and 13 days after inoculation with aphids. Three plants per variety were selected, and for each treatment (before inoculation or 7 or 13 days after inoculation with aphids), the first fully developed leaf at the top of each plant was collected. These leaves were then mixed to create a biological sample, with each treatment group containing three biological samples.

### 4.7. Quantitative Real-Time Reverse Transcription PCR Assay

The leaves (the first fully developed leaf at the top of each plant) of APL01 and Holly before aphid inoculation and on the 13th day after inoculation were collected for RT-qPCR assay. The leaves of three plants from each treatment (before inoculation or 13 days after inoculation with aphids) of each variety were collected and mixed into one biological sample, with each treatment containing three biological samples, resulting in 12 biological samples for RT-qPCR. Total RNA was isolated using a E.Z.N.A.^®^ Plant RNA Kit (Omega, Connecticut, USA), whereas RNA degradation and contamination were monitored on 1% agarose gels. The purity and concentration of the RNA were determined using a Biodrop µLite+ micro-volume spectrophotometer (Biodrop, New York, NY, USA). Primers for RT-qPCR were designed using the Primer 5 software (www.premierbiosoft.com/primerdesign/) (accessed on 15 October 2022) and synthesized by Tsingke Biotechnology Co. (Beijing, China). The gene-specific primers used for RT-qPCR assay are listed in Appendix A. Rapeseed *ACTIN* was used as an internal control, and triplicate quantitative assays were performed on each cDNA dilution using the Taq Pro Universal SYBR qPCR Master Mix (Vazyme, Nanjing, China) with the BIO-RAD CFX96 Real-Time PCR System (BIO-RAD, Hercules, CA, USA). The relative expression level of each gene was estimated using the 2^–ΔΔCt^ method [86].

## 5. Conclusions

In this study, through transcriptome, physiological, and gene expression analyses, it was revealed that chitinase activity, catalase activity, calcium signal transduction, and activation of systemic acquired resistance might be involved in aphid resistance in *B. napus*. The degree of inhibition of photosynthesis in plants under aphid stress directly determines the tolerance of *B. napus* to aphid stress. Furthermore, four promising candidate genes were screened from eight genes related to rapeseed response to biotic stress through RT-qPCR analysis of gene expression levels. These research findings represent an important step forward in understanding the resistance of rapeseed to aphid stress and provide a solid foundation for subsequent genetic analysis of traits related to rapeseed resistance to aphid stress and the cloning of genes responsible for this resistance.

## Figures and Tables

**Figure 1 plants-12-02855-f001:**
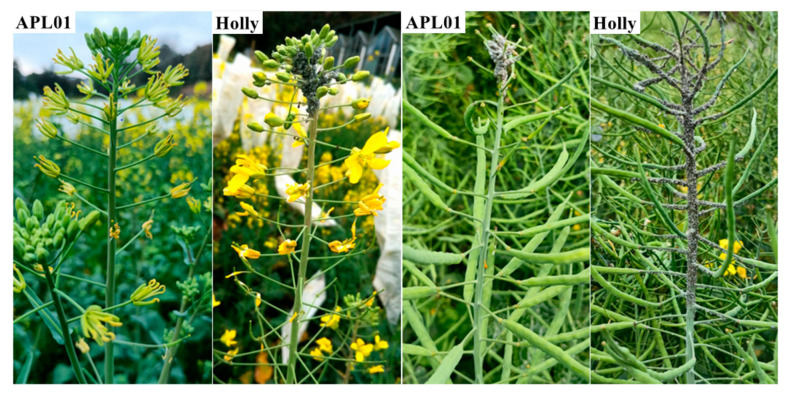
Aphid susceptibility of APL01 and Holly plants during the flowering and podding stages in the natural field. On 13 March and 25 April, respectively, observations were made on the susceptibility of plants to aphids during the flowering and podding stages, and one representative plant from each variety was selected for photography.

**Figure 2 plants-12-02855-f002:**
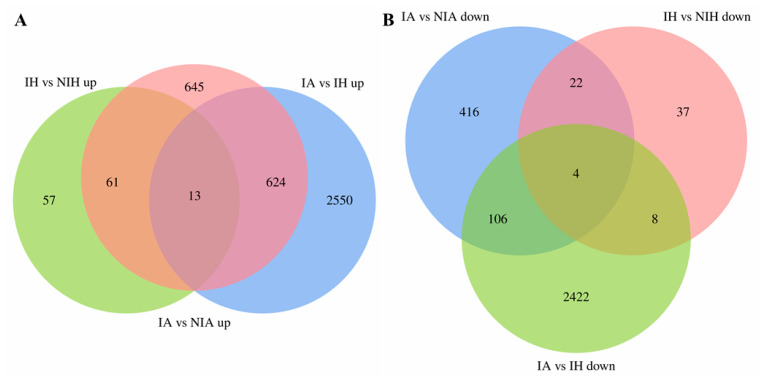
Venn analysis of DEGs (**A**) upregulated and (**B**) downregulated between IA vs. NIA, IH vs. NIH, and IA vs. IH groups. IA and IH, respectively, represent the young leaves of APL01 and Holly, which were inoculated with aphids for 13 days. NIA and NIH, respectively, represent the young leaves of APL01 and Holly that were not inoculated with aphids. “IA vs. NIA up/down” indicates the upregulated/downregulated DEGs identified in IA compared with NIA. “IH vs. NIH up/down” indicates the upregulated/downregulated DEGs identified in IH compared with NIH. “IA vs. IH up/down” indicates the upregulated/downregulated DEGs identified in IA compared with IH.

**Figure 3 plants-12-02855-f003:**
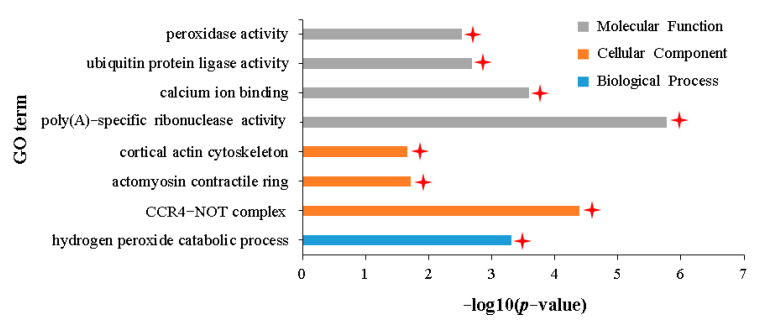
GO terms with specific functional descriptions significantly enriched by DEGs commonly upregulated in the IA vs. NIA and IH vs. NIH groups. Terms labeled with a red asterisk are associated with resistance to aphid stress. The definitions of IA, IH, NIA, and NIH are consistent with Figure 2. *p*-value represents the significance statistics of the enrichment of DEGs in a specific GO term, *p* < 0.05 or −log10 (*p*-value) > 1.3 indicates that the enrichment reaches a significant level.

**Figure 4 plants-12-02855-f004:**
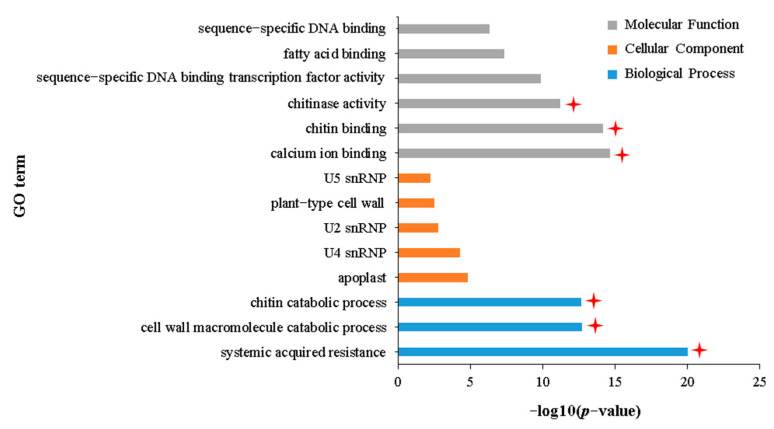
GO terms with specific functional descriptions significantly enriched by DEGs upregulated uniquely in the IA vs. NIA group. Terms labeled with a red asterisk are associated with resistance to aphid stress. The definitions of IA and NIA are consistent with Figure 2. The definition of *p*-value is consistent with Figure 3.

**Figure 5 plants-12-02855-f005:**
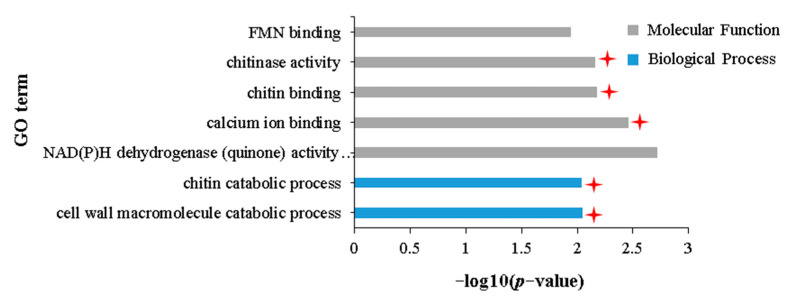
GO terms with specific functional descriptions significantly enriched by DEGs commonly upregulated in the IA vs. NIA, IH vs. NIH, and IA vs. IH groups. Terms labeled with a red asterisk are associated with resistance to aphid stress. The definitions of IA, IH, NIA, and NIH are consistent with Figure 2. The definition of *p*-value is consistent with Figure 3.

**Figure 6 plants-12-02855-f006:**
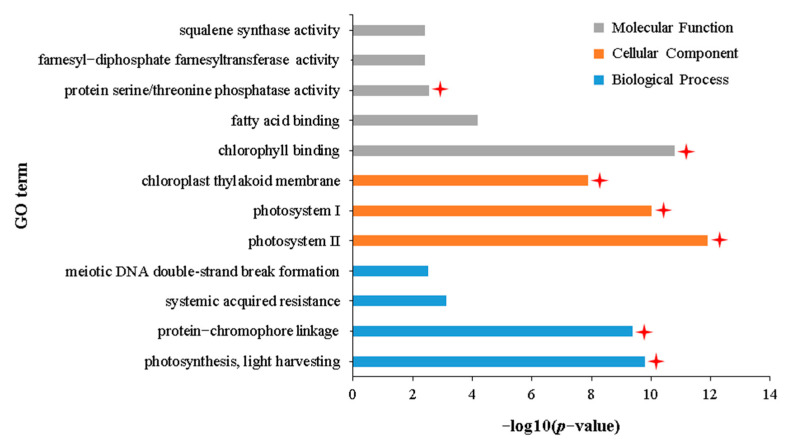
GO terms with specific functional descriptions significantly enriched by DEGs commonly downregulated in the IA vs. NIA and IH vs. NIH groups. Terms labeled with a red asterisk are associated with photosynthesis. The definitions of IA, IH, NIA, and NIH are consistent with Figure 2. The definition of *p*-value is consistent with Figure 3.

**Figure 7 plants-12-02855-f007:**
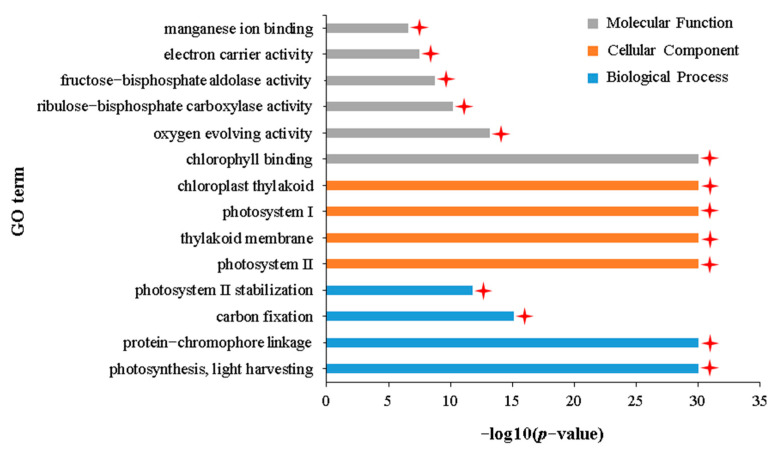
GO terms with specific functional descriptions significantly enriched by DEGs downregulated uniquely in the IA vs. NIA group. Terms labeled with a red asterisk are associated with photosynthesis. The definitions of IA and NIA are consistent with Figure 2. The definition of *p*-value is consistent with Figure 3.

**Figure 8 plants-12-02855-f008:**
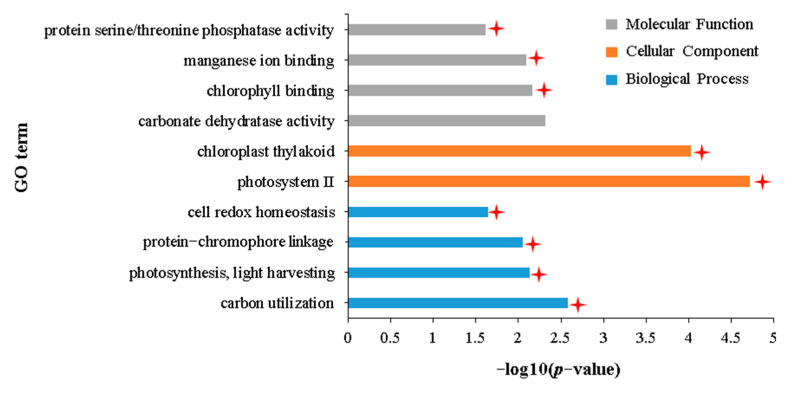
GO terms with specific functional descriptions significantly enriched by DEGs commonly downregulated in the IA vs. NIA, IH vs. NIH, and IA vs. IH groups. Terms labeled with a red asterisk are associated with photosynthesis. The definitions of IA, IH, NIA, and NIH are consistent with Figure 2. The definition of *p*-value is consistent with Figure 3.

**Figure 9 plants-12-02855-f009:**
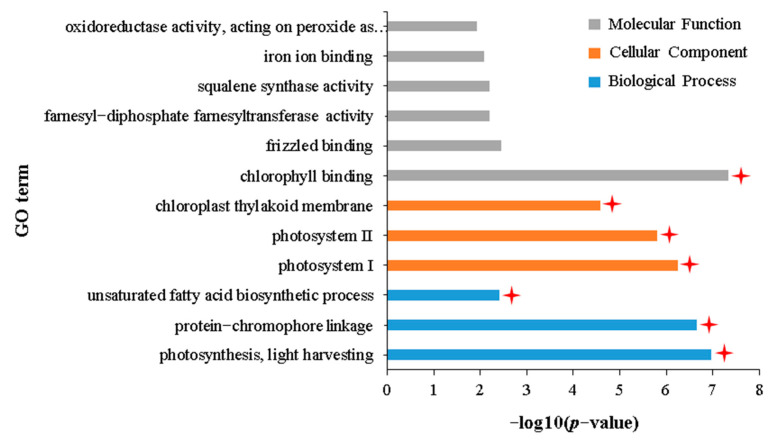
GO terms with specific functional descriptions significantly enriched by DEGs downregulated uniquely in the IH vs. NIH group. Terms labeled with a red asterisk are associated with photosynthesis. The definitions of IH and NIH are consistent with Figure 2. The definition of *p*-value is consistent with Figure 3.

**Figure 10 plants-12-02855-f010:**
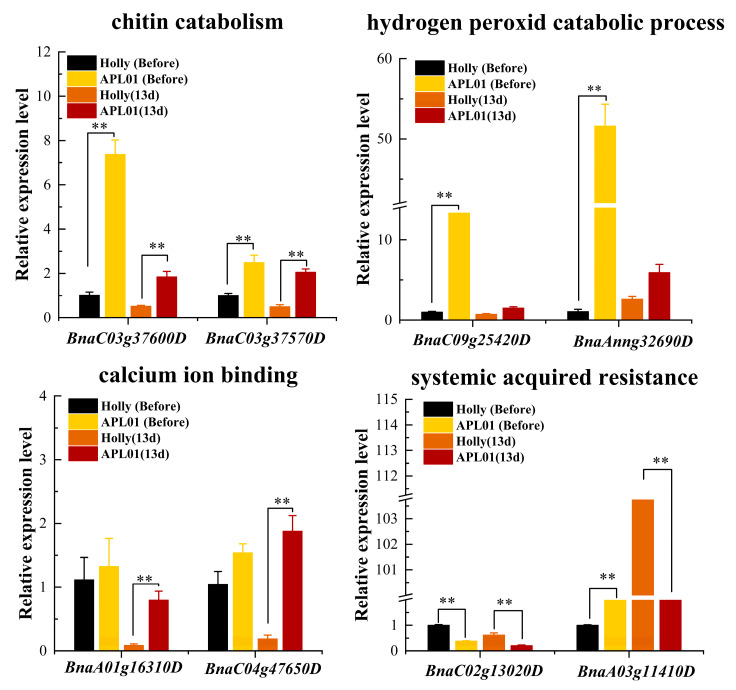
Expression levels of differentially expressed genes related to the response of rapeseed to aphid stress analyzed by RT-qPCR in APL01 and Holly plants before aphid inoculation and on the 13th day after inoculation. In the gene expression bar graph, the vertical axis represents the relative expression level of genes, and the horizontal axis represents genes. The black bars represent the gene expression level in the Holly leaves before aphid inoculation (labeled as “Before”), while the orange bars represent the gene expression level in the APL01 leaves before aphid inoculation (labeled as “Before”). The orange–red bars represent the gene expression level in the Holly leaves on the 13th day after aphid inoculation (labeled as “13d”), while the dark red bars represent the gene expression level in the APL01 leaves on the 13th day after aphid inoculation (labeled as “13d”). ** *p* < 0.01.

**Table 1 plants-12-02855-t001:** The number of aphids on APL01 and Holly plants at different times after inoculation.

Variety	7 d	10 d	13 d	16 d	Wilting Rate ^2^
APL01	11 ± 4.0 a	22 ± 8.96 a	59 ± 9.61 a	93 ± 13.23 a	66.7% a
Holly	8 ± 3.51 b	30 ± 11.15 b	66 ± 23.07 a	165 ± 73.50 b	28.6% b
Control ^1^	51 ± 39.88 c	109 ± 73.17 c	236 ± 167.94 b	398 ± 338.06 c	93.5% c

^1^ Chinese cabbage was used as a control. ^2^ Wilting rate of plants 25 days after inoculation with aphids. The unpaired *t*-test method was employed to analyze the significance of differences between pairwise data. Letters in the table represent the significance of differences between each pair of data.

**Table 2 plants-12-02855-t002:** Toxicity analysis of APL01 and Holly’s leaf extracts to aphids.

Variety	No. Aphids Inoculated	No. Dead Aphids	Mortality of Aphids
APL01	20	13.67 ± 2.08 a	68.3% a
Holly	20	12.67 ± 2.51 a	63.3% a
Control ^1^	20	1.33 ± 1.21 b	6.7% b

^1^ The extract solution without any powdered leaf added was used as a control. The unpaired *t*-test method was employed to analyze the significance of differences between pairwise data. Letters in the table represent the significance of differences between each pair of data.

**Table 3 plants-12-02855-t003:** The number of DEGs identified in different comparative groups.

Comparison Group	Up ^1^	Down ^2^
IA vs. NIA	1314	548
IH vs. NIH	131	71
IA vs. IH	3187	2540
NIA vs. NIH	2255	2217

^1^ The number of DEGs upregulated in a comparison group. ^2^ The number of DEGs downregulated in a comparison group. IA and IH, respectively, represent the young leaves of APL01 and Holly, which were inoculated with aphids for 13 days. NIA and NIH, respectively, represent the young leaves of APL01 and Holly that were not inoculated with aphids. “IA vs. NIA” refers to the number of DEGs identified in IA compared with NIA. “IH vs. NIH” refers to the number of DEGs identified in IH compared with NIH. “IA vs. IH” refers to the number of DEGs identified in IA compared with IH. “NIA vs. NIH” refers to the number of DEGs identified in NIA compared with NIH.

**Table 4 plants-12-02855-t004:** The number of upregulated DEGs related to the plant immune response to biotic stress in the IA vs. NIA and IH vs. NIH groups.

GO ID	Term	Annotated ^a^	Significant_A ^b^	Significant_H ^c^
GO:0042744	hydrogen peroxide catabolic process	258	11 (4.3%)	4 (1.6%)
GO:0005509	calcium ion binding	819	53 (6.5%)	5 (0.6%)
GO:0061630	ubiquitin protein ligase activity	360	8 (2.2%)	3 (0.8%)
GO:0004601	peroxidase activity	409	13 (3.2%)	4 (0.9%)
GO:0009627	systemic acquired resistance	97	21 (21.6%)	1 (1.0%)
GO:0016998	cell wall macromolecule catabolic process	59	14 (23.7%)	1 (1.7%)
GO:0006032	chitin catabolic process	60	14 (23.3%)	1 (1.7%)
GO:0008061	chitin binding	58	16 (27.6%)	1 (1.7%)
GO:0004568	chitinase activity	61	14 (22.9%)	1 (1.6%)
GO:0080032	methyl jasmonate esterase activity	11	0	2 (18.2%)
GO:0010333	terpene synthase activity	97	2 (2.1%)	2 (2.1%)
GO:0080031	methyl salicylate esterase activity	7	0	1 (14.3%)

^a^ The number of genes annotated with a GO term in the “Darmor-*bzh*” genome. ^b^ The number of upregulated DEGs annotated with a GO term in the IA vs. NIA group. ^c^ The number of upregulated DEGs annotated with a GO term in the IH vs. NIH group. The values in parentheses represent the percentage of DEG numbers in the total number of genes in the corresponding GO term.

**Table 5 plants-12-02855-t005:** The number of downregulated DEGs related to plant photosynthesis in the IA vs. NIA and IH vs. NIH groups.

GO ID	Term	Annotated ^1^	Significant_A ^2^	Significant_H ^3^
GO:0009765	photosynthesis, light harvesting	82	44 (53.7%)	9 (10.9%)
GO:0009523	photosystem II	177	81 (45.8%)	10 (5.6%)
GO:0009522	photosystem I	137	73 (53.3%)	9 (6.6%)
GO:0009535	chloroplast thylakoid membrane	362	72 (19.9%)	9 (2.5%)
GO:0016168	chlorophyll binding	90	44 (48.9%)	9 (10%)
GO:0030145	manganese ion binding	107	8 (7.5%)	1 (0.9%)
GO:0010242	oxygen evolving activity	11	7 (63.6%)	0
GO:0016984	ribulose-bisphosphate carboxylase activity	24	7 (29.2%)	0
GO:0004332	fructose-bisphosphate aldolase activity	21	6 (28.6%)	0
GO:0009055	electron carrier activity	549	17 (3.1%)	0
GO:0015977	carbon fixation	57	13 (22.8%)	0
GO:0042549	photosystem II stabilization	12	7 (58.3%)	0

^1^ The number of genes annotated with a GO term in the “Darmor-*bzh*” genome. ^2^ The number of downregulated DEGs annotated with a GO term in the IA vs. NIA group. ^3^ The number of downregulated DEGs annotated with a GO term in the IH vs. NIH group. The values in parentheses represent the percentage of DEG numbers in the total number of genes in the corresponding GO term.

**Table 6 plants-12-02855-t006:** Enzyme activities related to plant response to aphid stress at different times after inoculation with aphids.

Enzyme	Variety	Before	7 d	13 d
Peroxidase	APL01	81.62 ± 5.47 a	106.45 ± 3.90 a	214.05 ± 7.00 a
	Holly	60.46 ± 6.59 b	151.12 ± 19.85 b	128.57 ± 10.03 b
Catalase	APL01	99.76 ± 5.99 a	367.42 ± 5.35 a	330.61 ± 10.35 a
	Holly	174.34 ± 5.11 b	368.70 ± 4.10 a	249.89 ± 6.46 b
Chitinase	APL01	0.01505 ± 0.00001 a	0.01497 ± 0.00001 a	0.01493 ± 0.00001a
	Holly	0.01491 ± 0.00001 b	0.01490 ± 0.00001 a	0.01490 ± 0.00001a

“Before” refers to the time period before aphid inoculation. The unpaired *t*-test method was employed to analyze the significance of differences between pairwise data. Letters in the table represent the significance of differences between each pair of data.

**Table 7 plants-12-02855-t007:** APL01 and Holly’s photosynthesis-related parameters at different time points before and after aphid inoculation.

Index	Variety	Before	7d	13d
Net photosynthetic rate (μmol(CO_2_)m^−2^ s^−1^)	APL01	2.24 ± 1.08 a	7.75 ± 1.12 a	5.00 ± 1.03 a
	Holly	3.72 ± 1.34 a	13.43 ± 1.56 b	37.9 ± 1.87 b
Chlorophyll relative content (SPAD)	APL01	39.97 ± 2.69 a	40.77 ± 2.81 a	29.37 ± 2.54 a
	Holly	38.37 ± 2.10 a	36.03 ± 2.17 b	31.43 ± 2.79 a
Ribulose-bisphosphate carboxylase	APL01	85.51 ± 6.44 a	58.78 ± 4.43 a	51.45 ± 2.41 a
	Holly	47.16 ± 4.29 b	134.61 ± 6.52 b	330.79 ± 7.66 b
Fructose-bisphosphate aldolase	APL01	501.37 ± 82.45 a	292.42 ± 39.80 a	236.48 ± 28.39 a
	Holly	125.20 ± 22.41 b	267.68 ± 22.25 a	835.51 ± 21.01 b

“Before” refers to the time period before aphid inoculation. The unpaired *t*-test method was employed to analyze the significance of differences between pairwise data. Letters in the table represent the significance of differences between each pair of data.

## Data Availability

Not applicable.

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
