# Peer review of "Comparative Transcriptome Analysis Reveals the Molecular Basis of Brassica napus in Response to Aphid Stress"

_plants, 2023, doi:10.3390/plants12152855_

Round 1

Reviewer 1 Report

The main question addressed in this review is what the transcriptional differences between Brassica napus plants in response to aphid stress.  

This topic is relevant to the field. Understanding how plants response to insect infestations and what makes one plant resistant vs others that are susceptible is necessary in understanding the field.  

This is the first transcriptomic analysis, to my knowledge, that looks at these two lines of Brassica napus during aphid infections.  Other studies have used different plant species to observe similar things.  

The authors need to indicate where the field observations were made.  How many plants were observed and how they selected those plants to ensure there was not any bias selection.   The authors should further explain and hypothesize on how less aphids are linked to more wilting observed.   In figure 1, there needs to be an indication of dates the plants were observed.. Also, how were these plants chosen to be pictured in the publication.,  

 the conclusions are consistent with the evidence  

The references are appropriate

 Lines 112 through 115. Please adjust wording to make it more clear which abbreviation is which   Figure 4 - site source   Figure 10 - labels are to small.

Author Response

Reviewer 1:

Q1. The main question addressed in this review is what the transcriptional differences between Brassica napus plants in response to aphid stress.

Response: Thank you very much for your comments. According to the results of this study, the transcriptomic differences between the two Brassica napus varieties (APL01 and Holly) with different aphid susceptibility in response to aphid stress were mainly expressed in the expression of genes related to chitinase activity, Catalase activity, calcium signal transduction, systemic acquired resistance activation and photosynthesis.

Q2. This topic is relevant to the field. Understanding how plants response to insect infestations and what makes one plant resistant vs others that are susceptible is necessary in understanding the field.  

Response: Thank you very much for your comments.

Q3. This is the first transcriptomic analysis, to my knowledge, that looks at these two lines of Brassica napus during aphid infections. Other studies have used different plant species to observe similar things.  

Response: Thank you very much for your comments.

Q4. The authors need to indicate where the field observations were made. How many plants were observed and how they selected those plants to ensure there was not any bias selection. The authors should further explain and hypothesize on how less aphids are linked to more wilting observed. In figure 1, there needs to be an indication of dates the plants were observed.. Also, how were these plants chosen to be pictured in the publication.

Response: Thank you very much for your comments. Two B. napus varieties (APL01 and Holly) with significantly different aphid susceptibility and aphid density on plants were planted in two adjacent plots of the experimental rapeseed field of Guizhou University (Huaxi District, Guiyang City, Guizhou Province). Each plot contains 10 rows with 10 plants per row and 15 cm between the plants and 40 cm between the rows. All 100 plants of each variety were observed for aphid infestation during the flowering and podding stages in the field. Thus, there was not any bias selection.

In the early stage of aphid infestation, the proliferation rate of aphids on APL01 plants was significantly lower than that of Holly, while in the later stage, the withering rate of APL01 plants was much higher than that of Holly. This indicates that the inhibitory effect of APL01 on aphid proliferation is due to its antibiosis and antixenosis to aphids. However, APL01's ability to withstand aphid attacks is significantly weaker than that of Holly, indicating that APL01 has weaker tolerance to aphid stress than Holly. In summary, the mechanisms by which rapeseed varieties with different genetics respond to aphid stress vary. APL01 mainly relies on antibiosis and antixenosis to cope with aphid stress, while Holly mainly responds to aphid stress through antibiosis and tolerance. In the discussion section, we reinforced this viewpoint.

Observe the aphid susceptibility of APL01 and Holly plants during the flowering and podding stages on March 13 and April 25, 2023, respectively. Select a representative plant for each variety and take photos, as shown in Figure 1. We have added this information to the legend in Figure 1.

Q5. The conclusions are consistent with the evidence  

Response: Thank you very much for your comments.

Q6. The references are appropriate

Response: Thank you very much for your comments.

Q7. Lines 112 through 115. Please adjust wording to make it more clear which abbreviation is which

Response: Thank you very much for your comments. Based on your suggestion, we have reorganized these words and indicated what abbreviations IA, IH, NIA, and NIH represent respectively (Lines 132 through 134).

Q8. Figure 10 - labels are to small.

Response: Thank you very much for your comments. Based on your suggestion, we have re improved Figure 10.

Reviewer 2 Report

Moderate editing of English language is required.

Author Response

Reviewer 2:

The study of Li et al. seeks to understand plant responses and resistance to aphids in in two different varieties of Brassica napus. Through transcriptome, physiological and gene expression analyses the authors revealed main mechanisms displayed by both rapeseed varieties to aphid infection. It is a nice study with interesting results, which may be very useful for further agricultural. I have, however, some concerns, which should be addressed before publication:

Q1. First, the introduction felt a bit disorganized for the reader. Usually, the introduction should move from general statements to more specific details, ending with the study system. This kind of structure is not well organized here, and it is desirable. In the introduction it is also relevant to mention the novelty of the work.

Response:Thank you very much for your comments. Based on your suggestion, we have carefully read the introduction section of this manuscript and improved it.

Q2. Second, the methods are poorly described; several specific aspects are needed in order to validate the experiments and to replicate them in future.

Response:Thank you very much for your comments. Based on your suggestion, we have added a lot of descriptions about the experimental methods of this study, so that these experiments can be repeated and validated in the future.

Q3. Third, there is a confusion throughout the manuscript among the terms resistance, tolerance and defense. The authors use them as synonyms, and they are not. It is strictly necessary that these key concepts in the topic are properly used through the text.

Response:Thank you very much for pointing out this issue in the manuscript. Indeed, resistance and tolerance are two completely different concepts. There are three functions involved in plant-pest interactions [1]: antibiosis, antixenosis and tolerance, antibiosis, antixenosis. Among them, the first two belong to resistance, reflecting the active aggression of plants against insects. Tolerance reflects the plant's ability to withstand insect attacks. Thus, we have reorganized the entire text. This study mainly clarified that the low susceptible aphid rapeseed variety APL01 has stronger resistance to aphids than the high susceptible aphid variety Holly, but Holly has stronger tolerance to aphid stress than APL01. By comparing transcriptome analysis and physiological analysis, the transcriptome characteristics hidden in the differences in resistance and tolerance to aphid stress between APL01 and Holly were revealed. Further combining RT-PCR analysis, we screened four promising candidate genes from eight differentially expressed genes related to rapeseed resistance to biotic stress.

Additionally, I have some minor comments:

Q4. Line 74: Please show evidence (previous data) that both rapeseed varieties show indeed different levels of susceptibility to herbivores. And to which kind of herbivores? In general, plants, which are resistant to aphids, are susceptible to chewing insects and vice versa. Specific information regarding this is necessary to understand plant resistance strategies in both cultivars. Here, it is also important to mention the varieties the authors are working with.

Response:Thank you very much for your comments. The aphid susceptibility rates of two rapeseed varieties have been included as a result of this study in the results section. They are mainly affected by peach aphids in the field. Based on your suggestion, we have added this information at the end of the introduction. From flowering to podding, APL01 has an average of 25% susceptible to aphids compared with Holly's average of 93%.

Q5. Clear objectives/research questions and hypothesis are missing at the end of the introduction. Please include them.

Response:Thank you for pointing out this issue with this manuscript. We have added the purpose of this study at the end of the introduction. The main objectives are twofold: firstly, to clarify whether there is a difference in resistance to aphid stress between two B. napus varieties with different aphid susceptibility in the field, as well as the resistance categories of rapeseed to aphid stress (antibiosis or/and antixenosis or/and tolerance); the second is to clarify the transcriptome and physiological characteristics of B. napus in response to aphid stress.

Q6. Results: Before starting with the description of the results, a brief sentence describing what was done in each experiment is desirable.

Response:Thank you very much for your comments. In the results section of the first submitted manuscript, except for Chapter 2.1, other chapters contain a brief description of the experiments conducted in this section. In the revised manuscript, we have added an introduction to the experiments required in Chapter 2.1.

Q7. Line 99: Please mention which the blank control was.

Response:Thank you very much for your comments. The extract solution without any powdered leaf added was used as a control (i.e. blank extract). Although blank controls have been defined in the methodology section of this manuscript. Based on your feedback, we have added an explanation for the blank control at the corresponding position in Chapter 2.1.

Q8. Line 108: Results of Table 1 must be shown before Table 2. Change the order please.

Response:Thank you very much for your valuable suggestion. We have reversed the order of the results described in Table 1 and Table 2 in Chapter 2.1.

Q9. Table 6 and 7: 0d means before the before aphid inoculation, right? Please change it if it is so. 0 days after inoculation of aphids doesn’t make much sense.

Response:Thank you very much for your valuable suggestion. In this study, 0d means the before aphid inoculation. In the revised manuscript, we have changed '0d' to 'before inoculation'.

Q10. Table 6: Data of Rubisco and FBA should be shown in Table 7.

Response:Thank you very much for your valuable suggestion. In the revised manuscript, data of Rubisco and FBA have been shown in Table 7.

Q11. Figure 10: Please make it bigger. It was not possible to read the axis name, and thereby to interpret the results.

Response:Thank you very much for your valuable suggestion. In the revised manuscript, Figure 10 has been improved.

Q12. Line 428: What do the authors mean with tolerance? Resistance and tolerance to herbivores are 2 completely different terms. I have the idea that the authors tend to confuse both concepts.

Response: Thank you very much for pointing out this issue in the manuscript. Indeed, resistance and tolerance are two completely different concepts. There are three functions involved in plant-pest interactions [1]: antibiosis, antixenosis and tolerance, antibiosis, antixenosis. Among them, the first two belong to resistance, reflecting the active aggression of plants against insects. Tolerance reflects the plant's ability to withstand insect attacks. Thus, we have reorganized the entire text. This study mainly clarified that the low susceptible aphid rapeseed variety APL01 has stronger resistance to aphids than the high susceptible aphid variety Holly, but Holly has stronger tolerance to aphid stress than APL01. By comparing transcriptome analysis and physiological analysis, the transcriptome characteristics hidden in the differences in resistance and tolerance to aphid stress between APL01 and Holly were revealed. Further combining RT-PCR analysis, we screened four promising candidate genes from eight differentially expressed genes related to rapeseed resistance to biotic stress.

Q13. Line 533: How can the readers know that APL01 and Holly are differently susceptible to aphids? An additional table or figure is necessary to prove that.

Response: Thank you very much for your comments. From flowering to podding, APL01 has an average of 25% susceptible to aphids compared with Holly's average of 93% (Table S1). An additional table (Table S1) has been added to the Results and Methods section of the revised manuscript.

Q14. Inoculation experiment: Please describe in detailed how the experiment was carried out (factors and treatments, number of plant replicates, why 5 aphid per plant and not more?). If I understand well only three plants were used per treatment, this is low number for statistical analysis. How do the authors deal for this statistically? Statistical analyses are missing in the methods.

Response: Thank you very much for your comments. The aphid inoculation experiment involves two treatments (i.e. inoculation and non inoculation), with the only difference factor for each treatment being the variety (i.e. APL01 and Holly). The reproduction rate of peach aphids is very fast, and it only takes about 5 days to complete the first generation of reproduction. If more aphids are added to each plant, it will lead to a sharp increase in the number of aphids as the inoculation time prolongs, making it difficult to accurately count and unable to digitize the proliferation characteristics of aphids on different varieties. Inoculating five second instar weak aphids on each plant can ensure that the number of aphids on the plant can be accurately counted until the 16th day after inoculation. Therefore, in this study, only five aphids were inoculated on each plant. The above information is supplemented in Chapter 4.1

As you mentioned, containing only three plants per treatment seems to be insufficient for statistical analysis. More sample sizes can minimize experimental bias, and in future aphid inoculation experiments, we will appropriately increase the number of plants in each treatment. However, as far as this study is concerned, the bias of the aphid inoculation experiment is also within an acceptable range, and the statistical data is to some extent reliable. From the rapeseed varieties used in this study, APL01 and Holly are both genetically homozygous varieties, while in this study, three seedlings with 4 or 5 leaves of each variety were selected for aphid inoculation. So, the differences between seedlings are very small. Secondly, the aphids used in the inoculation experiment in this study were second instar peach aphids that had undergone more than three generations of reproduction and purification, and the differences between aphids have been controlled as much as possible. Finally, all plants inoculated with aphids are placed in an artificial climate box where temperature, light, and humidity are controllable and consistent, with small differences in the growth environment of individual plants. In summary, the data on aphid proliferation on APL01 and Holly obtained in this study is still reliable to a certain extent.

The calculation of mean and standard deviation, as well as the comparison between means, were carried out using SAS v9.4 software, and the paired t-test method was used for the significance analysis of differences between pairwise data.

Q15. Line 596: Please be more specific in the determination of photosynthesis and chlorophyll, how many leaves per plants were used, and which leaves? The same is valid for the other measurements (antioxidant enzymes, Rubisco and FAD).

Response: Thank you very much for your comments. In this study, the net photosynthetic rate, chlorophyll content, peroxidase activity, Catalase activity, Chitinase activity, ribulose-bisphosphate carboxylase activity, and fructose-bisphosphate aldolase activity of the first fully developed leaf were measured before and 7 and 13 days after aphid inoculation. In the revised manuscript, the above information was added to the methods section.

Q16. Discussion: this section is not well put in context of the literature. The current version of the discussion sounds more like a description of the results than an accurate discussion of the obtained results regarding what has been already described/found in the literature. I would also like to read like a future projection what it is expected to happen with resistant/susceptible cultivars to aphids in terms of resistance to other herbivores (chewing insect for example) and pathogens.

Response: Thank you very much for your comments. Based on your suggestion, we have improved the discussion section of this manuscript.

Considering the potential role of Chitinase activity in APL01 and Holly's resistance to aphids, it cannot be ruled out that APL01 and Holly may have a certain toxic effect on other stinging insects, but may not have a significant effect on chewing insects, because the Chitinase activity of these two varieties is not large enough.

References:

  1. Painter, R.H. Insect resistance in crop plants; The University Press of Kansas: Kansas, USA, 1951.

Reviewer 3 Report

The manuscript is a major contribution to the understanding of the defence response mechanisms in the economic highly important rapeseed. The manuscript includes a lot of analyses and data that can be used elsewhere later on. Nevertheless, I have some points I would like the authors to think about.

In the introduction a whole part is written about CORONATINE INSENSITIVE 1 (COI1), but later on through the analysis this gene is neither mentioned in DEG analysis nor in the RT PCR analysis. Why not? Didn’t you find it? If not, why not when it is so important?

I have a problem with the selection of the candidate genes for RT-qPCR. In Figure 10 there are 14 genes analysed, in chapter 2.8 you talk about 8 candidate genes and in the abstract 3 candidate genes are mentioned that have been analysed by RT-qPCR and in the discussion, I couldn’t clearly find which three candidate genes you mean. Could you please explain this more clearly.

Line 41: “Salicylic acid, abscisic acid, and jasmonic acid can also improve the tolerance of plants”. These are main players in defence response, thus, the wording is a little bit strange and too weak for these substances.

Line 44-46: ROS are products induced during the JA or SA cascade. So, this “similar to JA and SA” is misleading.

Line 66/67: “CORONATINE INSENSITIVE 1 (COI1) is currently the only gene cloned from rape-66 seed that is resistant to aphids”. Genes are not resistant! Correct would be, that COI1 is the only gene known to be associated with resistance of plants to aphids (or something like that).

Line 67/68: “jasmine receptor complex“. You mean jasmonate receptor complex?! Please. Look it up all over the manuscript, it is several times wrongly named.

Please, look through your wording throughout the text – there are more such unclear/incorrect formulations in the manuscript (e.g. line 543 ff “inoculated”)

Line 273: what is meant by “these four genes”? There are no specific genes mentioned here.

Line 438/439: This is a general statement. You analysed two varieties. I would therefore prefer that you relativize this statement by saying: These two analysed varieties of Brassica napus

The name of the chapter 4.3 is not adequate. It is besides the RNA sequencing also the bioinformatic analysis described.

Table 3 and Figure 2, 3, 4, 5 need a legend with an explanation of the used abbreviations. Legends of Tables and Figures must be written in a way that they can be understood on its own (without looking into the text). Please, keep this in mind and rewrite your legends of Tables and Figures.

Please, explain abbreviations at the first use. E.g. DAG is only explained on page 19, but used several times before that.

The resolution of Figure 10 is too low – the legends within the graphs can’t be read. Thus, I have no idea what the different colours stand for and therefore, can’t interpret the Figure 10.

Where are the legends to the supplementary files?

There is sometimes a weird wording and wrongly used terms. Please, correct this. Some examples are given in my review.

Author Response

Reviewer 3:

The manuscript is a major contribution to the understanding of the defence response mechanisms in the economic highly important rapeseed. The manuscript includes a lot of analyses and data that can be used elsewhere later on. Nevertheless, I have some points I would like the authors to think about.

Response: Thank you very much for your comments.

Q1. In the introduction a whole part is written about CORONATINE INSENSITIVE 1 (COI1), but later on through the analysis this gene is neither mentioned in DEG analysis nor in the RT PCR analysis. Why not? Didn’t you find it? If not, why not when it is so important?

Response: Thank you very much for your comments. CORONATINE INSENSITIVE 1 (COI1) is currently the only gene cloned from rapeseed that is resistant to aphids. In this study, the putative rapeseed COI1 gene (BnaA03g56600D) was not differentially expressed in the IA vs NIA, IH vs NIH, IA vs IH, and NIA vs NIH groups. It is speculated that the difference in resistance to aphid stress between APL01 and Holly is caused by differences in gene expression other than the COI1 gene. In fact, genetic studies have shown that plant resistance to aphids is jointly regulated by multiple loci and exhibits significant major gene effects. In this study, we identified that a large number of genes related to Chitinase activity, Catalase activity, calcium signal transduction, and activation of systemic acquired resistance were differentially expressed between APL01 and Holly, and these genes were potentially involved in differences in aphid resistance between APL01 and Holly.

Q2. I have a problem with the selection of the candidate genes for RT-qPCR. In Figure 10 there are 14 genes analysed, in chapter 2.8 you talk about 8 candidate genes and in the abstract 3 candidate genes are mentioned that have been analysed by RT-qPCR and in the discussion, I couldn’t clearly find which three candidate genes you mean. Could you please explain this more clearly.

Response: Thank you very much for your valuable suggestion. In this study, we aim to express that four promising candidate genes were screened from eight genes related to rapeseed resistance to aphid stress through RT-qPCR analysis of gene expression levels. Based on your suggestion, we have improved the language expression related to candidate gene screening in both the abstract and Chapter 2.8, and improved the quality of Figure 10, which now clearly reflects the meaning we want to express.

Q3. Line 41: “Salicylic acid, abscisic acid, and jasmonic acid can also improve the tolerance of plants”. These are main players in defence response, thus, the wording is a little bit strange and too weak for these substances.

Response: Thank you very much for your comments. According to your opinion, we have improved the description of the effects of salicylic acid, Abscisic acid and Jasmonic acid on the resistance of plants to aphids (Lines 40 through 44).

Q4. Line 44-46: ROS are products induced during the JA or SA cascade. So, this “similar to JA and SA” is misleading.

Response: Thank you very much for pointing out this inaccurate statement, and we have revised it.

Q5. Line 66/67: “CORONATINE INSENSITIVE 1 (COI1) is currently the only gene cloned from rape-66 seed that is resistant to aphids”. Genes are not resistant! Correct would be, that COI1 is the only gene known to be associated with resistance of plants to aphids (or something like that).

Response: Thank you very much for pointing out this inaccurate statement, and we have revised it.

Q6. Line 67/68: “jasmine receptor complex“. You mean jasmonate receptor complex?! Please. Look it up all over the manuscript, it is several times wrongly named.

Response: Thank you very much for pointing out this incorrect spelling. We have carefully reviewed the entire text and corrected such errors.

Q7. Please, look through your wording throughout the text – there are more such unclear/incorrect formulations in the manuscript (e.g. line 543 ff “inoculated”)

Response: Thank you very much for pointing out the wording issues in this manuscript. We have carefully reviewed the entire text and corrected such errors. Subsequently, we also sent the revised manuscript again to the International Science Editing (www.internationalscienceediting.com/) for language polishing.

Q8. Line 273: what is meant by “these four genes”? There are no specific genes mentioned here.

Response: Thank you very much for your comments. “these four genes” refer to four DEGs that were downregulated in the IA vs NIA, IH vs NIH, and IA vs IH groups. Among these three groups, we only identified four differentially expressed genes. In the text, we modify this statement to make it more accurate and clear.

Q9. Line 438/439: This is a general statement. You analysed two varieties. I would therefore prefer that you relativize this statement by saying: These two analysed varieties of Brassica napus…

Response: Thank you very much for your comments. Based on your suggestion, we have revised the corresponding statement.

Q10. The name of the chapter 4.3 is not adequate. It is besides the RNA sequencing also the bioinformatic analysis described.

Response: Thank you very much for your comments. As you mentioned, Chapter 4.3 also includes bioinformatics analysis, so we have optimized its name.

Q11. Table 3 and Figure 2, 3, 4, 5 need a legend with an explanation of the used abbreviations. Legends of Tables and Figures must be written in a way that they can be understood on its own (without looking into the text). Please, keep this in mind and rewrite your legends of Tables and Figures.

Response: Thank you very much for pointing out this defect. We have checked all the figures and tables in this manuscript and added legends with an explanation of the used abbreviations.

Q12. Please, explain abbreviations at the first use. E.g. DAG is only explained on page 19, but used several times before that.

Response: Thank you very much for pointing out this defect. We have checked all abbreviations in this manuscript and presented them in full when they first appeared.

Q13. The resolution of Figure 10 is too low – the legends within the graphs can’t be read. Thus, I have no idea what the different colours stand for and therefore, can’t interpret the Figure 10.

Response: Thank you very much for your comments. We have improved Figure 10 to make it more concise and clear.

Q14. Where are the legends to the supplementary files?

Response: We are very grateful that you pointed out this defect. We included all the supplementary information when we first submitted the manuscript to the editorial department, but this information is missing in the latest manuscript. We apologize for the issue and have added all supplementary information in the revised manuscript.

Round 2

Reviewer 2 Report

This is not an improved version of the manuscript. It seems that it has been done in a hurry. I highly recommend the authors to take time for the revision, and to resubmit a meticulous and well organized version of the manuscript. Overall, the manuscript still needs subtantial revision and improvement in the English writing. Please find more specific comments in the attached document.

Author Response

Reviewer 2:

There are some points, which are still not properly addressed by the authors. First, the introduction is still not well organized and structured. It is desirable to start with general statements and ending with the study system (not the other way around). Second, there is still a confusion throughout the manuscript among the terms resistance, tolerance and defense. Please correct this along the whole text. Additionally, there are several typing mistakes that need to be corrected.

Response: Thank you very much for your comments. We have carefully revised the introduction section of this manuscript. Starting from the introduction of rapeseed as an important oil crop worldwide, it is pointed out that aphids are the main pests that harm rapeseed production. Further introduce the current methods and existing problems of aphid control, thus highlighting the importance of cultivating aphid resistant varieties. In the second paragraph, we first briefly introduce the basic characteristics of plants harmed by aphids, and then introduce the secondary metabolites of phloem sap plants, the quality of phloem sap nutrients, and the role of phloem closure mechanisms in plant defense against aphids. At the same time, the role of two Plant hormone, Jasmonic acid and salicylic acid, in plant response to aphid stress was also introduced. In the third paragraph, we introduce the basic research progress on plant defense against aphid related traits, including genetic site mining and candidate gene cloning. Finally, the progress of genetic research on rapeseed defense against aphids was introduced, and the existing problems were pointed out. In the final paragraph, we briefly introduced all the materials and methods of this study, proposed the research objectives, and pointed out the research significance.

Secondly, we have reorganized the concepts of resistance, defense, and tolerance in this manuscript. This study conducted transcriptomic research on the leaves of two rapeseed varieties after inoculation and non-inoculation, and analyzed the changes in various stress-related indicators before and after inoculation at different time points for both varieties. These indicators include peroxidase activity, catalase activity, and photosynthesis. These research findings tend to reflect the transcriptomic and physiological characteristics of defense against aphids in the two rapeseed varieties. Of course, based on the changes in photosynthetic activity in response to aphid stress, the transcriptomic and physiological characteristics of the differences in tolerance to aphid stress between the two varieties have also been revealed.

Finally, we carefully read the entire text and made corrections to some errors, such as spelling and inaccurate expressions.

  • Line 86: This phrase is not clear.
  • Response:Thank you very much for your comments. We have improved this phrase.
  •  
  • Line 87: change clarify to evaluate
  • Response:Thank you very much for your comments. “clarify” has been changed to “evaluate”.
  •  
  • Line 89: change types to mechanisms or strategies
  • Response:Thank you very much for your comments. According to your suggestion, 'mechanisms' is used here.
  •  
  • Line 89: please change the word to clarify here (and different to evaluate)
  • Response:Thank you very much for your comments. We have changed the way of expression.
  •  
  • Line 89: delete as well as
  • Response:Thank you very much for your comments. “as well as” has been deleted.
  •  
  • Line 90: please change the word characteristics
  • Response:Thank you very much for your comments. The “characteristics” have been changed to “basis”.
  •  
  • Line 91: after the objectives, please add a prediction what it is expected to be found in the study
  • Response:Thank you very much for your comments. Based on your suggestion, a prediction what it is expected to be found in the study has been add
  •  
  • Line 93: this sentence is not clear
  • Response:Thank you very much for your comments. We have revised this sentence.
  •  
  • Line 380: CAT and POD are repeated
  • Response:Thank you very much for your comments. We have rechecked the entire text and found that "CAT" and "POD" appear here for the first time, so their full names have been written.
  •  
  • Lines 382-383: ‘before inoculation 0 days’: this sentence doesn’t make sense. Please change it throughout the whole text
  • Response:Thank you very much for your comments. In fact, our goal is to change '0 days' to 'before calculation'. But it may be due to its being presented in a traceable modification mode, resulting in "before calculation 0 days" being seen as a phrase. '0 days' has been deleted in the revised manuscript. Here, we have revised this sentence again.
  •  
  • Table 6 and 7: what does it mean 0days before inoculation? This sentence doesn’t make much sense. Please change it
  • Response:Thank you very much for your comments. In fact, our goal is to change '0d' to 'Before'. But it may be due to its being presented in a traceable modification mode, resulting in "0d Before" being seen as a phrase. '0d' has been deleted in the revised manuscript. And the definition of 'Before' is specified in the table annotation.
  •  
  • Line 416: FBA is repeated
  • Response:Thank you very much for your comments. We have rechecked the entire text and found that "FBA" appears here for the first time, so its full name has been written.
  •  
  • Line 417: Table 67??
  • Response:Thank you very much for your comments. In fact, in the phrase 'Table 67', '6' has been deleted from the revised manuscript. In fact, our goal is to change 'Table 6' to 'Table 7'. But it may be due to its being presented in a traceable modification mode, resulting in "Table 67" being seen as a phrase. '6' has been deleted in the revised manuscript.
  •  
  • Figure 10: legend and axis information is still too small
  • Response:Thank you very much for your comments. We have revised the axis information and legend of Figure 10.
  •  
  • Line 622: each row was. Change it please
  • Response:Thank you very much for your comments. We have corrected this mistake.
  •  
  • Methods: Plant material section: there is a confusion here, there are red and blue text, but there is not a fluidity and connection between both.
  • Response:Thank you very much for your comments. The reason why red and blue text appears in the revised manuscript is because different co authors have edited the manuscript. As you said, this can easily make people feel confused, so we have revised the entire text to address this issue.
  •  
  • Line 640: undrer??
  • Response:Thank you very much for your comments. In fact, our goal is to change 'undre' to 'under'. But it may be due to its being presented in a traceable modification mode, resulting in "undrer" being seen as a word. 'undre' has been changed to “under” in the revised manuscript.
  •  
  • Line 643: cacse??
  • Response:Thank you very much for your comments. In fact, our goal is to change 'cacse' to 'case'. But it may be due to its being presented in a traceable modification mode, resulting in "cacse" being seen as a word. 'cacse' has been changed to “case” in the revised manuscript.
  •  
  • The methods should be written in past sentence (not present). Change it along the text
  • Response:Thank you very much for your comments. Based on your suggestions, we have made changes to the grammar in the methodology section.
  •  
  • Lines 645-649: this sentence is not clear
  • Response:Thank you very much for your comments. We have revised this sentence. Plants that were not inoculated with aphids were also covered with the same isolation cover to control for differences in environmental factors affecting plant growth, such as light, temperature, and humidity, between the inoculation and non-inoculation treatments. All plants, regardless of treatment, were cultured in an artificial climate box with controlled conditions (light 25℃, 16 h; dark 20℃, 8 h; relative humidity 80%).
  •  
  • Why did the authors use pair-t-tests for the analyses? I don’t think this the appropriate analysis
  • Response:Thank you very much for your comments. As you mentioned, the paired t-test is not applicable in this study, so we used the unpaired t-test method. However, we accidentally omitted the "un" when writing "unpaired". Thank you very much for pointing out this mistake. We have carefully read the entire text and made corrections of such errors.
  •  
  • Line 710: of each of each?
  • Response:Thank you very much for your comments. "of each of each" has been corrected to "of each".
  •  
  • Lines 711-714: this sentence is not clear
  • Response:Thank you very much for your comments. We have revised this sentence to read: “The net photosynthetic rate of the first fully developed leaf at the top of each selected plant was measured before inoculation, as well as 7 and 13 days after inoculation with aphids. Additionally, the relative chlorophyll content at three positions in the distal third of the first fully developed leaf at the top of each selected plant was measured between 9:00 and 11:00 am before inoculation and 7 and 13 days after inoculation with aphids.”
  •  
  • In general, the methods are still not well described. Please be careful describing this section in order to get reproducibility.
  • Response:Thank you very much for your comments. In the previous two versions of the manuscript, there were unclear descriptions of some experimental methods in this study. In this revised version, we have reorganized the unclear sentences to make them more comprehensible.
  •  
  • Line 737: experiments were not developed before inoculation and at 0 days. Please be careful with these kinds of mistakes.
  • Response:Thank you very much for your comments. In fact, in this study, the activities of POD, CAT, Rubisco, and FBA in the leaves were measured before aphid inoculation and on the 7th and 13th days after inoculation.
  •  
  • Discussion: I still don’t see this section well put in context of the literature. Please see my comments in the first-round revision.
  • Response:Thank you very much for your comments. According to your suggestions, in the discussion section, we have minimized the repetitive descriptions of our research results while incorporating the cited literature reports to draw reasonable inferences. Additionally, we have included a discussion on the potential correlation between host plant adaptations for chewing-type insect defense and piercing-sucking type insect defense. From the perspective of plant defenses against these two types of insects, it is undeniable that there is a certain degree of consistency between these two defense mechanisms.
  •  
  • In general, this version is not improved. Please take the time to make it properly.

        Response: Thank you very much for your comments. Based on your and another reviewer's suggestions, we have carefully revised this manuscript.

Reviewer 3 Report

You really improved the manuscript. I still have only some small remarks:

Q2: Please, have again a look at chapter 2.8. It seems that the track changes are not correct – some words that, I guess, have been deleted by you are still there.

That, unfortunately, seems to have happened throughout the text (e.g. explanation of abbreviations are now given, but the original text is still there).

Q9: The sentence you revised (line 504 – 506) must be rewritten – it seems that two authors made changes and the result is a not readable sentence.

Q13: The legends within the graphs of Figure 10 are still not readable. Thus, I still have no idea what the different colours stand for and can’t interpret the Figure 10.

Author Response

Reviewer 3:

You really improved the manuscript. I still have only some small remarks:

Q2: Please, have again a look at chapter 2.8. It seems that the track changes are not correct – some words that, I guess, have been deleted by you are still there.

Response: Thank you very much for your comments. Perhaps due to significant differences in the format of the manuscript we downloaded from the Plants journal system from our first submission, we were unable to fully identify the issues you pointed out in the review report. Based on your suggestion, we have carefully reviewed Chapter 2.8 and corrected any errors or unclear information.

That, unfortunately, seems to have happened throughout the text (e.g. explanation of abbreviations are now given, but the original text is still there).

Response: Thank you very much for your comments. Perhaps due to the swapping of the order between the method section and the results section after the manuscript was submitted to the Plants journal, some abbreviations lacked full names when they first appeared. In this revised manuscript, we have checked all abbreviations to ensure that they are represented in full when they first appear in the main text.

Q9: The sentence you revised (line 504 – 506) must be rewritten – it seems that two authors made changes and the result is a not readable sentence.

Response: Thank you very much for your comments. As you mentioned, this sentence was modified by two co authors, resulting in the text appearing in different colors, which makes it look confusing. In the revised manuscript, we have rewritten this sentence.

Q13: The legends within the graphs of Figure 10 are still not readable. Thus, I still have no idea what the different colours stand for and can’t interpret the Figure 10.

Response: Thank you very much for your comments. We have revised the axis information and legend of Figure 10.
